# Essential function of transmembrane transcription factor MYRF in promoting transcription of miRNA *lin-4* during *C. elegans* development

Zhimin Xu, Zhao Wang, Lifang Wang*, Yingchuan B Qi*

School of Life Science and Technology, ShanghaiTech University, Shanghai, China

**Abstract** Precise developmental timing control is essential for organism formation and function, but its mechanisms are unclear. In *C. elegans*, the microRNA *lin-4* critically regulates developmental timing by post-transcriptionally downregulating the larval-stage-fate controller LIN-14. However, the mechanisms triggering the activation of *lin-4* expression toward the end of the first larval stage remain unknown. We demonstrate that the transmembrane transcription factor MYRF-1 is necessary for *lin-4* activation. MYRF-1 is initially localized on the cell membrane, and its increased cleavage and nuclear accumulation coincide with *lin-4* expression timing. MYRF-1 regulates *lin-4* expression cell-autonomously and hyperactive MYRF-1 can prematurely drive *lin-4* expression in embryos and young first-stage larvae. The tandem *lin-4* promoter DNA recruits MYRF-1$^{GFP}$ to form visible loci in the nucleus, suggesting that MYRF-1 directly binds to the *lin-4* promoter. Our findings identify a crucial link in understanding developmental timing regulation and establish MYRF-1 as a key regulator of *lin-4* expression.

*For correspondence:
lifangwang06@gmail.com (LW);
qiyc@shanghaitech.edu.cn
(YBQ);
qiyc@shanghaitech.edu.cn (YBQ)

**Competing interest:** The authors declare that no competing interests exist.

## eLife assessment

The microRNA *lin-4*, originally discovered in *C. elegans*, has a key role in controlling developmental timing across species, but how its expression is developmentally regulated is poorly understood. Here, the authors provide **convincing** evidence that two MYRF transcription factors are essential positive regulators of *lin-4* during early *C. elegans* larval development. These results provide **important** insight into the molecular control of developmental timing that could have significant implications for understanding these processes in more complex systems.

## Introduction

It is during the embryonic stage that the basic form and pattern of organisms are mostly established. However, the majority of growth in animals occurs during post-embryonic stages (***Wolpert et al., 2019***). Some extreme examples are seen in insects undergoing metamorphosis and vertebrate amphibians (***Shi, 2013***). In many cases, there is often a considerable increase in body size, but profound transitions take place, such as neural circuitry remodeling throughout juvenile years and sexual maturation (***Huttenlocher, 1979***; ***Sun and Hobert, 2023***; ***Truman, 1990***). Undoubtedly, the genetic program determines how different species grow to different characteristic sizes and forms, yet such mechanisms remain largely unknown. One key question is how developmental timing is controlled.

When hatching out of their egg shells, the nematode *C. elegans* exhibits a miniature yet grossly similar form to adults, and demonstrates a set of behavioral abilities including sensory response,

locomotion, and learning (*White et al., 1976*; *White et al., 1986*; *Wood, 1988*). They go through four larval stages (L1 to L4) before becoming fertile adults, and the transition between two consecutive larval stages is marked by a molting event. Changes in the temporal pattern of development are best exemplified by stage-specific cell division and differentiation of blast cells, as well as stage-specific epidermal cuticle formation fate. Such stage-specific patterns can be skipped or reiterated in so-called heterochronic mutants, and *lin-4* and *lin-14* are the two that have been studied in detail (*Ambros and Horvitz, 1984*; *Ambros and Moss, 1994*; *Lee et al., 1993*; *Rougvie and Moss, 2013*; *Ruvkun and Giusto, 1989*; *Slack and Ruvkun, 1997*; *Wightman et al., 1993*). *lin-4* mutants exhibit complex cell lineage defects, including reiterated larval stage 1-specific cell division in some progenitors, while *lin-14* mutants exhibit precocious patterning where stage-specific events are skipped. The studies of these mutants led to the discovery of the first microRNA regulatory pathway as follows: The ubiquitously expressed nuclear factor LIN-14 promotes L1 patterns and suppresses progression to L2. The microRNA *lin-4* is upregulated during late L1, and suppresses LIN-14 post-transcriptionally, consequently initiating progression to the L2 pattern (*Feinbaum and Ambros, 1999*; *Olsen and Ambros, 1999*). *lin-4* - LIN-14 pair not only controls the division pattern of blast cells but also controls the maturation and plasticity of the neural circuit during L1-L2 transition (*Hallam and Jin, 1998*; *Howell et al., 2015*; *Sun and Hobert, 2021*); *lin-4* is also involved in other diverse biological processes (*Ambros and Ruvkun, 2018*). The vertebrate ortholog of *lin-4*, known as *miR-125*, has been found to promote neuronal differentiation and maturation (*Akerblom et al., 2014*; *Boissart et al., 2012*).

What remains a mystery is the factors that trigger the expression of the *lin-4* microRNA during mid-late L1. It is known that the coding sequence of *lin-4* is embedded within an intron of a host gene, and its transcription uses its own promoter and bound Pol II complexes (*Bracht et al., 2010*; *Feinbaum and Ambros, 1999*; *Lee et al., 1993*). To date, no essential, positive regulator of *lin-4* transcription has been conclusively identified. However, one significant negative regulator has been reported: FLYWCH transcriptional factors suppress *lin-4* transcription during embryonic stages, a suppression that extends into late embryogenesis (*Ow et al., 2008*). Notably, FLYWCH mutants fail to progress to normal hatched larvae, implying that FLYWCH plays a pivotal role and may have additional functions beyond its role in suppressing *lin-4*. This discovery hints at the existence of a poorly understood mechanism governing the transition from embryos to larvae. Equally enigmatic is the process of activating *lin-4* transcription during the transition from L1 to L2.

The onset of *lin-4* expression in late L1 is likely linked to the nutritional state. *C. elegans* needs to feed to initiate the post-embryonic developmental programs of the L1 stage. When newly hatched animals encounter an environment without food, they enter into a diapause state in which development is suspended, and they become more resistant to environmental stress. In the case of epidermal blast cell division, the cycling inhibitors are promoted by more activated FOXO transcription factor DAF-16 due to starvation, which suppresses the blast's division (*Baugh and Hu, 2020*; *Baugh and Sternberg, 2006*). During L1 diapause, *lin-4* expression is suppressed, but this suppression is largely independent of *daf-16* (*Baugh and Sternberg, 2006*), suggesting that *lin-4* expression onset during mid-late L1 under nutrient-rich environment is not a result of simple attenuation of *daf-16* activity. Given that the *lin-4* expression initiates in late L1, it is reasonable to deduce that merely providing food is inadequate to induce *lin-4* expression. Instead, *lin-4* expression likely results from the interplay between the internal nutritional status and developmental progress.

Another unique developmental event that occurs at late L1 is the synaptic rewiring of DD motor neurons (*Hallam and Jin, 1998*; *Mizumoto et al., 2023*; *White et al., 1978*). The process enables a structural and functional switch between the dendritic and axonal domains of DDs without an obvious transition in gross neuronal morphology. We previously identified MYRF-1 and MYRF-2, transmembrane transcription factors, as essential positive regulators of synaptic rewiring in DDs (*Meng et al., 2017*). While *myrf-1* null mutants show an arrest at the end of the L1 stage, they only display a mild deficiency in synaptic rewiring. In contrast, *myrf-2* null mutants do not exhibit any notable growth defects, including in synaptic rewiring. However, the *myrf-1; myrf-2* double null mutants demonstrate a significant impairment in synaptic rewiring, indicating that both genes collaboratively drive this process. The gain-of-function analysis by overexpressing either *myrf-1* or *myrf-2* is sufficient to advance the onset timing of synaptic rewiring. Notably, a specific *myrf-1* mutation, *ju1121* G274R, identified in the initial screen, shows a severe deficiency in synaptic rewiring, similar to the *myrf-1; myrf-2* double mutants (*Meng et al., 2017*). Molecular analysis suggests that the *ju1121*(G274R) mutation impairs MYRF-1's

DNA binding ability and also interferes with MYRF-2's function, resulting in a negative interference effect. It is important to note that the synaptic rewiring defect does not directly correlate with larval arrest, as both *myrf-1; myrf-2* double mutants and *myrf-1(ju1121)*, despite their significant synaptic rewiring deficiencies, arrest at a later stage (L2) than *myrf-1* single mutants. These findings collectively underscore the predominant role of *myrf-1* in regulating development in *C. elegans*.

MYRF is conserved across the metazoan and indispensable for animal development in both invertebrates and vertebrates (*Bujalka et al., 2013*; *Emery et al., 2009*; *Li et al., 2013*; *Russel et al., 2011*). In mice, *Myrf* is well-recognized for its role in promoting myelination postnatally and maintaining it in adults (*Emery et al., 2009*; *Koenning et al., 2012*). Additionally, *Myrf* is essential for early embryonic development in mice, although its specific functions during this stage remain uncharacterized. Haplo-insufficiency in human *MYRF* leads to *MYRF*-Related Cardiac Urogenital Syndrome (*Kaplan et al., 2022*). MYRF exhibits several distinctive domain features that facilitate its intricate processing and activation (*Bujalka et al., 2013*; *Kim et al., 2017*; *Li et al., 2013*). The full-length protein initially integrates into the membrane, and undergoes trimerization via its intramolecular chaperone domain, which triggers self-catalyzing cleavage. This process releases the N-terminal MYRF in its trimeric form, allowing it to enter the nucleus for transcriptional regulation. When the catalytic dyad responsible for self-cleavage is mutated, MYRF-1(S483A K488A, GFP) (in *myrf-1(syb1487)*) remains constantly on the cell membrane, leading to mutant animals phenotypically similar to *myrf-1* null mutants (*Xia et al., 2021*). Trimerization is not only a prerequisite for self-cleavage but is also essential for the proper functioning of the N-terminal MYRF. When not correctly assembled and processed into a trimer, the N-terminal MYRF, in its monomeric form, lacks functional sufficiency (*Kim et al., 2017*; *Muth et al., 2016*). This is further evidenced by the phenotypic resemblance of mutants expressing the N-terminal alone, such as *myrf-1(syb1491, 1–482, GFP)* and *myrf-1(syb1468, 1–656, GFP)*, to *myrf-1* null mutants in terms of larval arrest and synaptic rewiring (*Xia et al., 2021*). Notably, these mutant MYRF proteins do properly translocate into the nucleus. Despite its pivotal role in animal development, the regulation of MYRF's processing during development, as well as MYRF's transcriptional targets (other than myelin-related genes), remains poorly defined.

In our previous research, we observed that in *C. elegans*, the MYRF-1 protein localizes to the cell membrane during early and mid-L1 stages but increasingly undergoes self-cleavage towards late L1, triggered by as-yet-unidentified signaling mechanisms (*Xia et al., 2021*). The trafficking of MYRF to the cell membrane relies on a second transmembrane protein with leucine-rich repeat domains, PAN-1 (*Xia et al., 2021*). When PAN-1 is absent, MYRF fails to reach its intended destination and undergoes degradation in the endoplasmic reticulum (ER). It is the extracellular region of MYRF that facilitates its interaction with PAN-1. When this region (701-931) is deleted (in *myrf-1(syb1313)*), truncated MYRF-1 (1–700, GFP) remains trapped in the ER instead of being trafficked to the cell membrane, resulting in deficient processing of MYRF (*Xia et al., 2021*). This emphasizes the critical importance of the cell membrane location for MYRF's normal processing. However, the ER-located MYRF-1 (1–700, GFP) (in *myrf-1(syb1313)*) can be processed to a very limited extent. This processing is not governed by developmental timing and leads to discordant, premature development in certain tissues while the whole animal of *myrf-1(syb1313, 1–700, GFP)* arrests during the second larval stage (*Xia et al., 2021*). Therefore, it is essential for MYRF to be trafficked to the cell membrane for proper cleavage. This process involves the interactions between the vesicular luminal (or extracellular) regions of MYRF and PAN-1.

To date, despite the MYRF-1's essential role in regulating larval development in *C. elegans*, the functional targets of MYRF-1 remain elusive. In the course of studying the genetic interaction between *myrf-1* and *lin-14*, we discovered that MYRF-1 is required for *lin-4* expression. We present data demonstrating that MYRF-1 is an essential, cell-autonomous driver of *lin-4* expression.

## Results

### Nuclear accumulation of N-MYRF-1 coincides with the activation of *lin-4*

The accumulation of *lin-4* microRNA occurs during mid to late L1, primarily due to the activation of *lin-4* primary RNA transcription, as supported by multiple studies (*Bracht et al., 2010*; *Ow et al., 2008*; *Figure 1A and C*). We confirmed their findings by observing a P*lin-4::GFP* (*maIs134*) reporter generated by the Ambros group (*Ow et al., 2008*), consisting of 2.4 kb of DNA sequences upstream of the mature *lin-4* fused to GFP. After its on-set, *lin-4* expression appears to be constitutive and

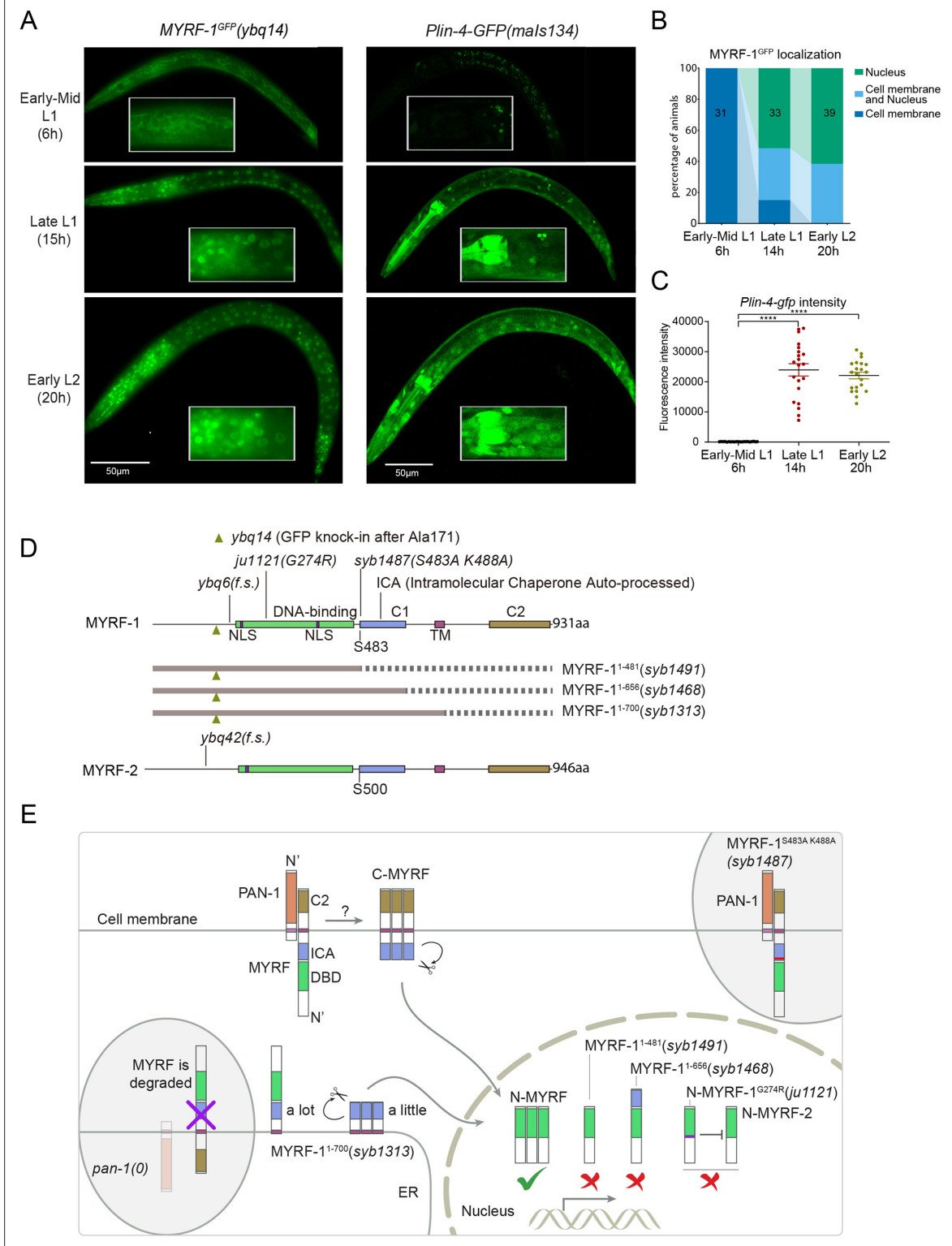

**Figure 1.** The nuclear accumulation of N-MYRF-1 coincides with the induction of *lin-4* in developmental timing. (**A**) Nuclear localization of GFP::MYRF-1 is increased in late L1, coinciding with the expression of *Plin-4-GFP*. GFP, endogenously inserted at MYRF-1 Ala171, labels both full-length MYRF-1 and post-cleaved N-MYRF-1. *Plin-4-GFP(maIs134)* is a transcriptional reporter of *lin-4*, carrying a 2.4 kb sequence upstream of the *lin-4* gene that drives GFP. While GFP::MYRF-1 is initially localized at the cell membrane in early-mid L1 (6 post-hatch hours), it becomes enriched in the nucleus towards late L1 (15 post-hatch hours). *Plin-4-GFP* is barely detected in early L1 but is upregulated in late L1. The insert shows a zoomed-in view of the framed area, covering part of the pharynx. (**B**) Quantification of animals showing a particular pattern of GFP::MYRF-1 (as shown in (**A**)) at various stages. The number

*Figure 1 continued on next page*

*Figure 1 continued*

of animals analyzed is indicated on each bar. (*Figure 1—source data 1*). (**C**) The fluorescence intensity of the *lin-4* transcriptional reporter (as shown in (**A**)) was quantified and presented as mean ± SEM (t-test, ****p<0.0001). (*Figure 1—source data 2*). (**D**) This illustration presents the gene structures of *myrf-1* and *myrf-2*, along with the mutations we investigated. The self-cleavage sites are S483 in *myrf-1* and S500 in *myrf-2*. The *myrf-1(ybq14)* variant involves GFP insertion at Ala171. Both *myrf-1(ybq6)* and *myrf-2(ybq42)* have indel mutations that lead to frameshifts, rendering them functionally null. The *myrf-1(ju1121, G274R)* variant contains a missense mutation and results in a phenotype similar to the *myrf-1; myrf-2* double mutants. The *myrf-1(syb1487, S483A K488A)* variant carries modifications at two catalytic residues that are important for self-cleavage. The variants *myrf-1(syb1491, 1–482)*, *myrf-1(syb1468, 1–656)*, and *myrf-1(syb1313, 1–700)* represent in-frame deletions, each tagged with GFP at Ala171.E. This model outlines the trafficking pathways, subcellular locations, processing activities, and regulatory roles of MYRF, as well as the impact of dysfunctional MYRF mutants in *C. elegans*. Initially, the full-length MYRF protein is situated in the endoplasmic reticulum (ER), where it binds with PAN-1, a protein with a leucine-rich repeat (LRR) domain. This interaction is critical for MYRF's movement to the cell membrane. Without PAN-1, MYRF is degraded within the ER. The process of MYRF trimerization and its subsequent cleavage at the cell membrane is tightly timed, though the exact mechanisms behind this are yet to be fully understood. Once cleaved, the N-terminal part of MYRF (N-MYRF) moves to the nucleus, playing a key role in specific developmental processes. The diagram also illustrates the functional effects of MYRF variants resulting from genetic modifications. Mutating MYRF-1's conserved catalytic dyad residues to S483A K488A results in the mutant protein's persistent retention on cell membranes, causing an arrest phenotype at the end of the first larval stage, similar to null *myrf-1* mutants. Mutants expressing only the N-terminal segment (MYRF-1 (1–481)) or including the ICA domain (MYRF-1 (1–656)) are constantly present in the nucleus. These variants, however, show developmental issues akin to null *myrf-1* mutants. The MYRF-1(G274R) mutation disrupts DNA binding and also inhibits MYRF-2 function through direct interaction, resulting in a phenotype resembling *myrf-1; myrf-2* double mutants. Removing the vesicular luminal regions in MYRF-1 primarily causes the MYRF-1 (1–700) variant to stay in the ER without appropriate processing, although a small portion does undergo cleavage. Animals with the *myrf-1 (1–700)* mutation typically experience developmental arrest during the L1 and L2 stages, showing a combination of both premature and delayed cellular development.

The online version of this article includes the following source data for figure 1:

**Source data 1.** The quantification of animals exhibiting a specific pattern of GFP::MYRF-1 at various stages, as illustrated in *Figure 1B*.

**Source data 2.** The detailed statistical analysis of the fluorescence intensity for the lin-4 transcriptional reporter, as illustrated in *Figure 1C*.

ubiquitous throughout the larval stages (see discussion). MYRF-1 is also expressed broadly in larvae, but unlike *lin-4*, MYRF-1 transcription is active in both embryos and early L1, with an increase in transcription observed towards late L1 (*Meng et al., 2017*). However, the activity of MYRF-1 as a transcription factor is determined by the presence of N-MYRF-1 in the nucleus (*Xia et al., 2021*). Initially, full-length MYRF-1 localizes to the cell membrane during early L1, and only during mid to late L1 is the processing of MYRF-1 cleavage increased, resulting in an elevated amount of N-MYRF-1 being released and shuttled into the nucleus (*Figure 1A and B*). Therefore, the nuclear accumulation of N-MYRF-1 coincides with the induction of *lin-4* both temporally and spatially.

## MYRF-1 is required for *lin-4* expression

We previously reported that two loss-of-function deletion mutants of *myrf-1* (*syb1491* and *syb1468*) (*Figure 1D and E*) exhibit phenotypic similarities to putative *myrf-1* null mutants, particularly in their arresting stages (the end of L1) and mild synaptic rewiring deficits (*Xia et al., 2021*). The *lin-4* transcription reporter (*maIs134*) is not activated in these mutants (*Figure 2—figure supplement 1*), indicating that *myrf-1* plays a crucial role in promoting *lin-4* transcription. In another allele, *myrf-1(ju1121 G274R)*, the MYRF-1 mutant protein not only loses its DNA binding capability but also negatively interferes with its close paralogue MYRF-2 (*Figure 1D and E*; *Meng et al., 2017*). Consequently, *myrf-1(ju1121)* displays phenotypic resemblance to double mutants of *myrf-1* and *myrf-2*, exhibiting severe synaptic rewiring blockage and arrest during L2 (*Meng et al., 2017*), one stage later than *myrf-1* single mutants. Our analysis reveals that the *maIs134* reporter fails to be expressed in *myrf-1(ju1121)* (*Figure 2A and B*), with the latest stage being during L2-3 molting. Therefore, based on the 2.4 kb promoter-reporter analysis, the activation of *lin-4* at late L1 is dependent on the presence of MYRF.

To confirm the crucial role of MYRF in the activation of *lin-4* transcription, we investigated how *myrf* influences *lin-4* transcription by employing a reporter system with a nucleus-localized mScarlet protein, endogenously inserted at the *lin-4* locus (*umn84*), wherein the reporter open reading frame replaced the primary RNA sequence of *lin-4*. Our observations revealed that mScarlet signals were not detected in early L1 larvae (*Figure 2C–F*; *Figure 2—figure supplement 2*). These signals markedly increase during the late L1 stage and exhibit even stronger intensity in the early L2 stage. The mScarlet signals exhibit a marked reduction in the putative null mutant *myrf-1(ybq6)* (*Figure 1D and E*). Intriguingly, in the putative null *myrf-2(ybq42)* mutants, there is a noticeable trend towards increased mScarlet signals, although this increase does not reach statistical significance (*Figure 2C*

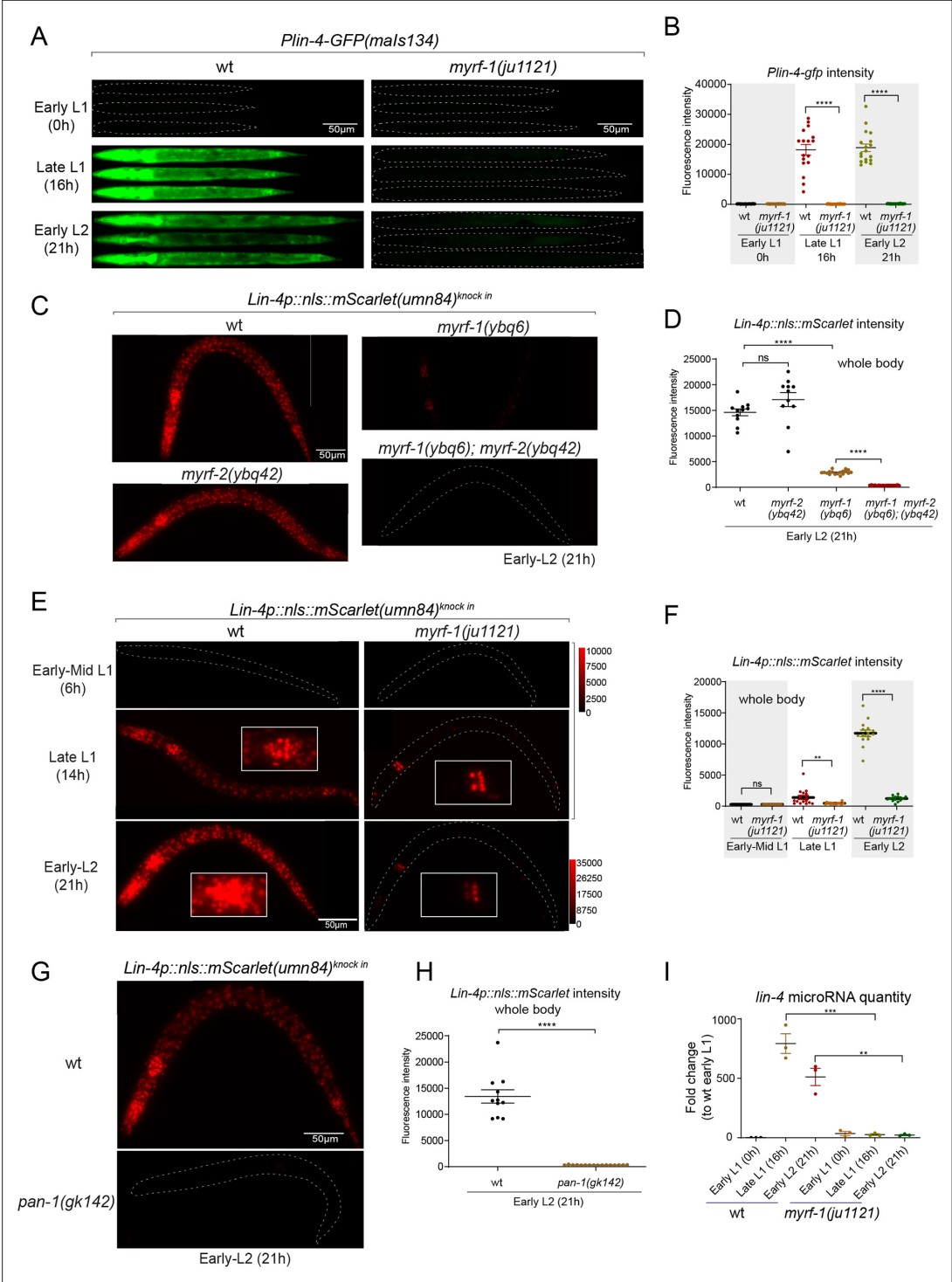

**Figure 2.** MYRF is required for *lin-4* expression during late L1. (**A**) *Plin-4-GFP(maIs134)* is not expressed in *myrf-1(ju1121)*. The expression of the *lin-4* transcriptional reporter in wild-type and *myrf-1(ju1121)* animals was examined at the early L1 (0 hr), late L1 (16 h), and early L2 stages (21 hr). (**B**) The fluorescence intensity of the *lin-4* transcriptional reporter (as shown in (**A**)) was quantified and presented as mean ± SEM (t-test, ****p<0.0001). (***Figure 2—source data 1***). (**C**) The expression of the endogenous *lin-4* transcriptional reporter, *lin-4p::nls::mScarlet(umn84)*, is substantially reduced in *myrf-1(ybq6)* mutants. In contrast, there is no significant change in its expression in *myrf-2(ybq42)* mutants. Importantly, this reporter is not activated in the *myrf-1; myrf-2* double mutants. The weak expression of the reporter can be visualized by decreasing the maximum display range. Such representation is available in (***Figure 2—figure supplement 3***). (**D**) The fluorescence intensity of the *lin-4* reporter, as illustrated in (**C**), was quantified and is presented as mean ± SEM. Statistical significance was determined using a t-test (****p<0.0001; ns, not significant). The term 'whole body' refers to the region of interest (ROI) encompassing the entire body of the animal. This definition is consistently applied throughout this figure(**E**) The *lin-4* reporter (*umn84*)

*Figure 2 continued on next page*

*Figure 2 continued*

is not activated in *myrf-1(ju1121)* mutants, with the exception of a few pharyngeal nuclei. The expression of this reporter in both wild-type and *myrf-1(ju1121)* animals was examined at various developmental stages: early to mid L1 (6 hr), late L1 (14 hr), and early L2 (21 hr). Due to the strong expression of the reporter in the wild-type, the maximum display range has been increased to prevent signal saturation. The adjusted display range is indicated in the intensity scale adjacent to the images. (F) The fluorescence intensity of the *lin-4* reporter, as depicted in (E), was quantified and is presented as mean ± SEM. Statistical significance was assessed using a t-test (****p<0.0001). (*Figure 2—source data 3*). For quantification of the *lin-4* reporter in head neurons and pharyngeal cells, refer to (*Figure 2—figure supplement 3*). (G) The *lin-4* reporter (*umn84*) is completely inactive in *pan-1(gk142)* mutants. These mutants and their wild-type controls were examined at the early L2 stage (21 hr). Images with a decreased maximum display range are available in (*Figure 2—figure supplement 3*). (H) The fluorescence intensity of the *lin-4* reporter, as shown in (G), was quantified and is presented as mean ± SEM. Statistical significance was determined using a t-test (****P<0.0001). (*Figure 2—source data 4*). (I) The abundance of mature *lin-4* miRNAs in wild-type and *myrf-1(ju1121)* animals was examined at the early L1 (0 hr), late L1 (16 hr), and early L2 stages (21 h) using qPCR analysis with probes specifically detecting *lin-4* microRNA. Each data point represents relative fold change of each sample compared to the wild-type Early L1 sample within one set of experiment. The data represent three biological replicates. Statistics used t-test. *p<0.05, ***p<0.001. (*Figure 2—source data 5*).

The online version of this article includes the following source data and figure supplement(s) for figure 2:

**Source data 1.** The statistical analysis of the fluorescence intensity of the lin-4 transcriptional reporter (maIs134) in wild-type and myrf-1(ju1121) mutants, as illustrated in *Figure 2B*.

**Source data 2.** The statistical analysis of the fluorescence intensity of lin-4p::nls::mScarlet in wild-type, myrf-1(ybq6), myrf-2(ybq42), and myrf-1(ybq6); myrf-2(ybq42) double mutants, as illustrated in *Figure 2D*.

**Source data 3.** The statistical analysis of the fluorescence intensity of lin-4p::nls::mScarlet in wild-type and myrf-1(ju1121) mutants at the mid L1 (6 hr), late L1 (16 hr), and early L2 (21 hr) stages, as illustrated in *Figure 2F*.

**Source data 4.** The statistical analysis of the fluorescence intensity of lin-4p::nls::mScarlet in wild type and pan-1(gk142) mutant , as illustrated in *Figure 2H*.

**Source data 5.** The abundance of mature lin-4 miRNAs in wild-type and myrf-1(ju1121) animals at the early L1 (0 hr), late L1 (16 hr), and early L2 (21 hr) stages using qPCR analysis, as illustrated in *Figure 2I*.

**Figure supplement 1.** *Plin-4-GFP* expression in *myrf-1* loss of function mutants.

**Figure supplement 1—source data 1.** The statistical analysis of the fluorescence intensity of the lin-4 transcriptional reporter (maIs134) in wild-type, myrf-1(syb1491), and myrf-1(syb1468) animals at the L2 stage (24 hr), as illustrated in *Figure 2—figure supplement 1B*.

**Figure supplement 2.** The expression of *lin-4* endogenous reporter throughout larval stages.

**Figure supplement 3.** MYRF-1 is required for *lin-4* expression.

**Figure supplement 3—source data 1.** The statistical analysis of the fluorescence intensity of lin-4p::nls::mScarlet in head neurons and pharyngeal cells in wild-type and myrf-1(ju1121) mutants, as illustrated in *Figure 2—figure supplement 3D*.

**Figure supplement 4.** MYRF-1 is required for *lin-4* expression throughout the larval stages.

**Figure supplement 4—source data 1.** The detailed statistical analysis of the fluorescence intensity of the lin-4 transcriptional reporter (maIs134) in larval stages L2, L3, and L4 animals treated with or without 4 mM K-NAA for 12 hours, as illustrated in *Figure 2—figure supplement 4B*.

*and D*). When the *myrf-1(ybq6)* mutation is combined with *myrf-2(ybq42)*, the mScarlet signals disappear under comparable detection conditions (*Figure 2C and D*). Only under a greatly reduced maximum display range can some signals be inconsistently observed in the intestines of the double mutants (*Figure 2—figure supplement 3*). Given that the distribution pattern of these residual signals differs significantly from the normal *lin-4* expression pattern (e.g. strong in the pharynx, epidermis, and neurons), it is questionable whether these signals truly reflect the regulation of *lin-4*. These results suggest that both *myrf-1* and *myrf-2* contribute to promoting *lin-4* expression, with *myrf-1* playing a predominant role and *myrf-2* a minor role.

We analyzed this endogenously tagged *lin-4* expression reporter (*umn84*) in *myrf-1(ju1121)* mutants (*Figure 2E and F*). We find that the mScarlet signals were largely absent throughout the body during the late L1 stage (14 hr), except for an interesting presence in 8–9 nuclei in the posterior bulb of the pharynx, with intensity comparable to that observed in the wild-type counterparts (*Figure 2E and F*; *Figure 2—figure supplement 3*). The overall absence of *lin-4* expression persisted in *myrf-1(ju1121)* mutants during early L2 stage (21 hr), except for the aforementioned subset of pharyngeal nuclei where the mScarlet signals increased, although the ascent was much weaker compared to the wild-type controls.

We previously discovered that the trafficking of MYRF-1 and MYRF-2 to the cell membrane depends on the transmembrane, leucine-rich repeat domain-containing protein PAN-1 (*Xia et al., 2021*). In the absence of PAN-1, MYRF is unstable and undergoes degradation in the ER. *pan-1(gk142)* deletion

mutants exhibit severe synaptic rewiring blockage in DD neurons and undergo progressive larval arrest, never progressing beyond the L3 stage (*Gao et al., 2012*; *Gissendanner and Kelley, 2013*; *Xia et al., 2021*). Interestingly, the larval arrest in *pan-1* mutants occurs at a notably later stage than *myrf-1*; *myrf-2* double mutants. We find that the scarlet *lin-4* expression reporter (*umn84*) is not activated in *pan-1(gk142)* mutants (*Figure 2G and H*; *Figure 2—figure supplement 3B*), consistent with the notion that MYRF is inactive without PAN-1.

We carried out qPCR analysis using probes specifically targeting *lin-4* microRNA to examine endogenous *lin-4* expression in wild-type and *myrf-1(ju1121)* mutants, and observed a significant reduction in the levels of mature *lin-4* microRNA in *myrf-1(ju1121)* mutants (*Figure 2I*). This decrease in mature *lin-4* microRNA has been further confirmed through microRNA sequencing analysis (Figure 7A). These findings, combined with the analysis of endogenous *lin-4* reporter, provide compelling evidence supporting the critical role of MYRF in the induction of *lin-4* during the late L1 stage.

To investigate if MYRF-1 is continuously required for *lin-4* transcription after its initial activation, we used the auxin-inducible degradation (AID) system (*Zhang et al., 2015*). We combined the ubiquitously expressed F-box protein TIR1 with degron-tagged MYRF-1 to acutely deplete MYRF-1 protein. The degradation was induced by treating animals at L2, L3, and L4 stages with an K-NAA (1-Naphthaleneacetic acid potassium salt) solution, an auxin analog. Our results showed a significant reduction in the *lin-4* transcription reporter signals in animals of all tested stages within 10 hr post-treatment (*Figure 2—figure supplement 4*). This demonstrates that MYRF-1 is necessary for *lin-4* expression throughout the larval stages.

## Sustained high level of LIN-14 protein in *myrf-1* mutants

The LIN-14 protein is typically present in embryos and early L1 but is downregulated as development progresses. Loss of the microRNA *lin-4* leads to sustained high levels of LIN-14 protein throughout

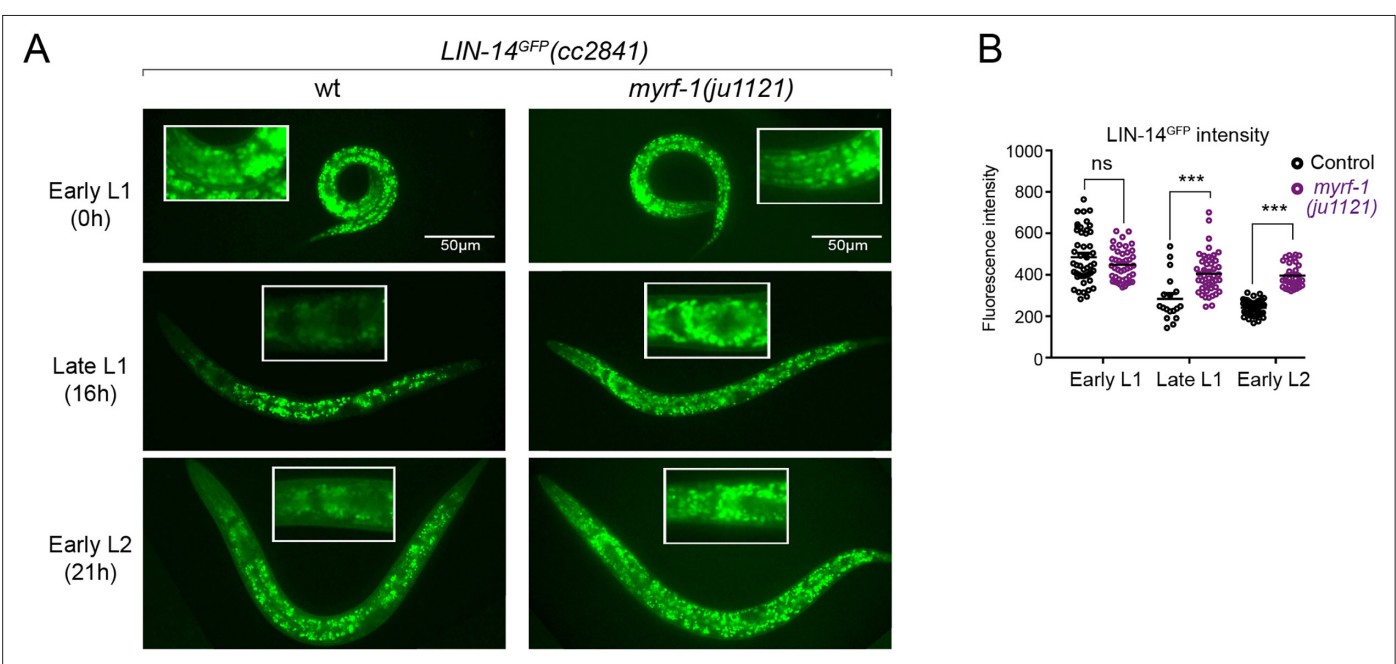

**Figure 3.** There is a sustained high level of LIN-14 protein in *myrf-1* mutants. (**A**). Expression of LIN-14::GFP(*cc2841*) in wild-type and *myrf-1(ju1121)* at the early L1 (0 hr), late L1 (16 hr), and early L2 (21 hr) stages. GFP was endogenously tagged at the LIN-14 C-terminus. LIN-14::GFP is bright in early L1 and downregulated in late L1. LIN-14::GFP is not affected by *myrf-1(ju1121)* at early L1 but is significantly brighter than wild-type control at late L1 and L2. (**B**). The fluorescence intensity for LIN-14::GFP (as shown in **A**) was measured and presented as mean ± SEM (t-test, ns: not significant, p>0.05; ***p<0.001). Each data point represents the mean intensity of the head region in an individual animal. The head region was selected due to its low autofluorescence background. (*Figure 3—source data 1*).

The online version of this article includes the following source data for figure 3:

**Source data 1.** The detailed statistical analysis of the fluorescence intensity for LIN-14::GFP in wild type and myrf-1(ju1121) mutant, as illustrated in *Figure 3B*.

larval development (*Wightman et al., 1993*). To investigate how *myrf-1* mutations may affect LIN-14, we examined the signals of endogenously tagged LIN-14::GFP in *myrf-1(ju1121)* mutants (*Arribere et al., 2014*). The LIN-14::GFP signal remains bright and shows no signs of decreasing at late L1 and beyond in *myrf-1(ju1121)* mutants (*Figure 3A and B*), consistent with the low levels of *lin-4* microRNA observed in the mutants.

## MYRF-1 regulates *lin-4* expression cell-autonomously

It is currently unclear that how MYRF specifically regulates the larval development. Identifying *lin-4* as a potential transcriptional target of MYRF-1, we aimed to investigate how MYRF-1 may regulate *lin-4* expression. One key question is whether MYRF-1 promotes *lin-4* expression in terminal target cells or whether MYRF-1 acts in discrete tissues to control unidentified systemic signals that subsequently lead to *lin-4* expression.

We conducted tissue-specific rescue experiments by expressing MYRF-1 under the *myo-3* promoter, which is specific to body wall muscles, in *myrf-1(ju1121)* mutants. We observed a significant induction of *lin-4* transcription in body wall muscles but not in other tissues (*Figure 4A*). We also used the epidermis-specific promoter of *dpy-7* for another MYRF-1 rescue experiment, which resulted in the appearance of *lin-4* transcription reporter signals only in the epidermis. Notably, the reporter GFP signals were absent in the seam cells, a group of specialized epidermal cells embedded in the syncytium epidermal cell. This aligns with the observation that the short promoter of *dpy-7*, employed in this experiment, remains inactive in seam cells (*Gilleard et al., 1997*), therefore excluding the induction of *lin-4* in these cells. These results demonstrate that MYRF regulates *lin-4* transcription autonomously within specific cells.

One concern over the lack of *lin-4* activation in *myrf-1* loss of function mutants is that it may be due to overall developmental arrest during L2. To investigate this possibility, we generated a conditional allele of *myrf-1*LoxP(*ybq98*) using CRISPR-Cas9 editing. We then combined an epidermis-specific Cre-expressing transgene with the *myrf-1*LoxP allele in animals (*Figure 4B and C*). Our analysis showed that the *lin-4* transcription reporter signals in the epidermis are lost in animals with the combined Cre transgene and *myrf-1*loxP allele, while the signals were still present in all surrounding tissues. Interestingly, animals with dual alleles develop into adults without apparent abnormalities (*Figure 4D and E*). However, these adults display a consistent body elongation, a phenotype similar to that observed in *lin-4* loss of function mutants. Additionally, all of them die within two days due to internal organ spillage caused by vulva bursting and internal larvae hatching. These results provide further evidence that MYRF-1 promotes *lin-4* activation cell autonomously and suggest that MYRF-1 acts directly in terminal tissues to regulate cell development.

## Hyperactive form of MYRF-1 drives premature expression of *lin-4*

Given that both the induction of *lin-4* and the cleavage of MYRF at the cell membrane happen within a specific time window, we investigated whether a gain of function in MYRF-1 alone is adequate to modify the onset timing of *lin-4*. Because of the intricate nature of regulated MYRF-1 cleavage, over-expressing full-length MYRF-1 by transgene does not enhance its transcriptional activity effectively, as excess MYRF-1 is unable to traffic to the cell membrane or undergo adequate cleavage (*Meng et al., 2017*). Overexpressing N-MYRF-1 alone is also insufficient as it unlikely to forms trimers efficiently. In another study, we have identified elements of the cleavage mechanism, which we will report in another manuscript. Using this information, we created a truncated form of MYRF-1 (deleting 601–650) that is expected to bypass the need for cell-membrane trafficking and circumvent the developmental signals that control cleavage. We expected that overexpressing this MYRF-1 variant would produce abundant N-MYRF-1 in trimeric form and enhance its endogenous function.

In this experiment, we used the endogenously mScarlet-tagged line (*umn84*) to report the transcription activity of *lin-4*. mScarlet expression is not detected in embryos and early L1 larvae (*Figure 5A*). We find that overexpressing the hyperactive MYRF-1(Δ601–650) driven by the ubiquitous promoter *rpl-28* is detrimental to larval development, as all F1 progenies with the transgene arrest at early larval stages. Strikingly, expressing this hyperactive MYRF-1 variant causes the expression of *lin-4* transcription reporter in both embryos and early L1 larvae, indicating that MYRF-1 alone is sufficient to activate *lin-4* transcription (*Figure 5B*). Given that *lin-4* transcription is suppressed by FLYWCH during

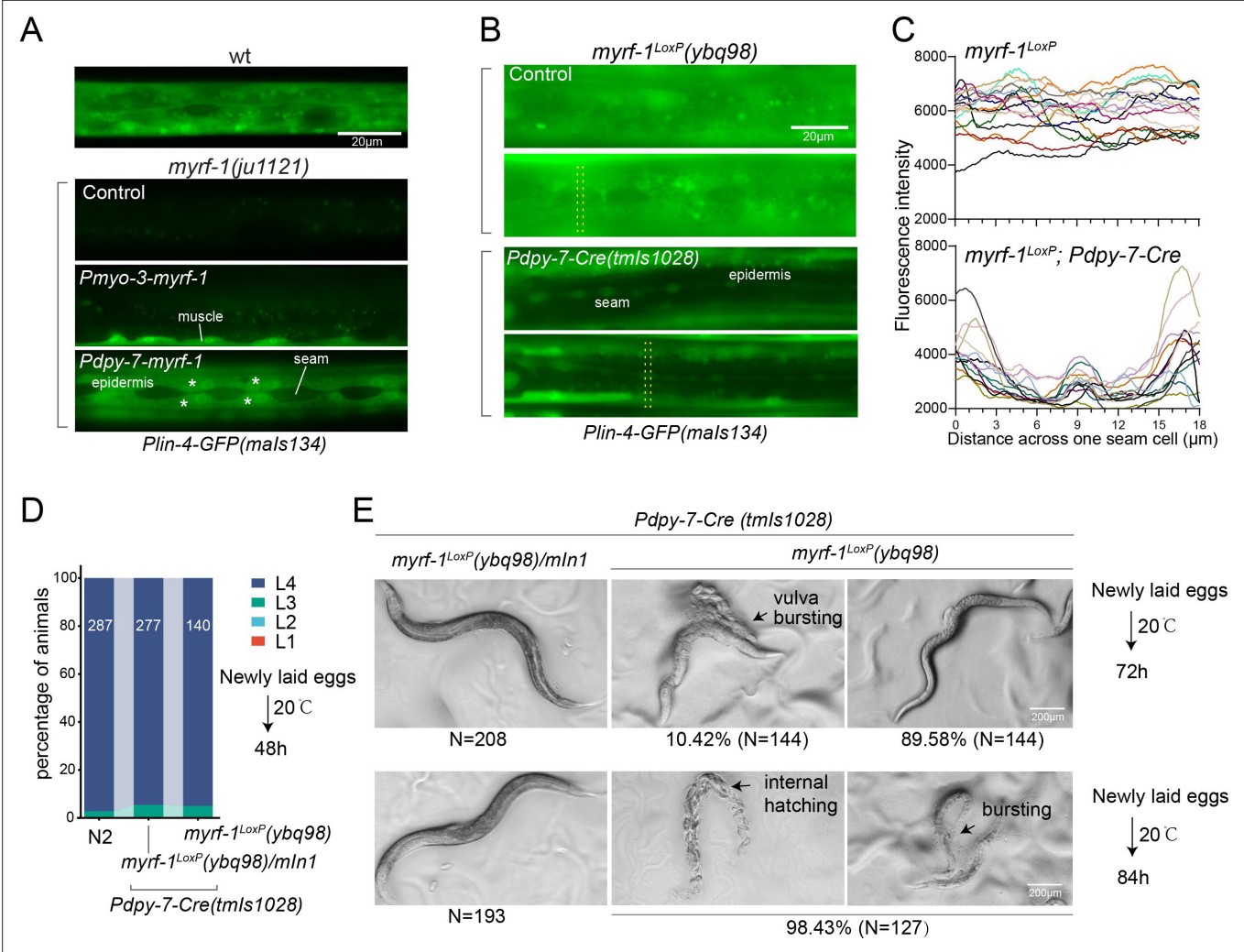

**Figure 4.** MYRF-1 is sufficient to drive *lin-4* expression in a cell-autonomous manner. (**A**). Genetic rescue of MYRF-1 in *myrf-1*(*ju1121*) using tissue-specific promoters. *Plin-4-GFP(maIs134)* signals are observed specifically in body wall muscles and epidermis (asterisk) of *myrf-1*(*ju1121*) carrying transgene *Pmyo-3-myrf-1* and *Pdpy-7-myrf-1*, respectively, while no detectable *Plin-4-GFP* is observed in L2 of *myrf-1*(*ju1121*). (**B**). Tissue-specific ablation of *myrf-1* in the epidermis. *myrf-1*LoxP(*ybq98*) combined with *Pdpy-7-NLS::Cre(tmIs1028)* caused loss or drastic decrease of *Plin-4-GFP(maIs134)* in the epidermis, while signals were detected in other tissues. Representative images of L2 (24 h) animals are shown. (**C**). Fluorescence intensity data, illustrated in (**B**), are presented for a transversely oriented ROI (Region of Interest) bar centered on a single seam cell. Each line in the data corresponds to the signal from an individual animal. (*Figure 4—source data 1*).(**D**). Effects of ablating *myrf-1* in epidermis using *Cre-LoxP*. Assessments of larval growth in *myrf-1*LoxP(*ybq98*); *Pdpy-7-Cre(tmIs1028)* compared to control animals show that larval stage development does not exhibit obvious defects. (*Figure 4—source data 2*). (**E**) By day 1, adult *myrf-1*LoxP(*ybq98*); *Pdpy-7-Cre(tmIs1028)* animals exhibit internal organ spillage through vulva bursting in about 10% of day-1 adults, while others' bodies exhibit elongation (similar to *lin-4* mutants). By day 2, nearly all *myrf-1* ablated animals are dead, either due to bursting or internal hatching.

The online version of this article includes the following source data for figure 4:

**Source data 1.** The detailed statistical analysis of the fluorescence intensity of the lin-4 transcriptional reporter(maIs134) in wild-type and myrf-1(ybq98); Pdpy-7-Cre(tmIs1028) animals, as illustrated in *Figure 4C*.

**Source data 2.** Assessments of developmental stages for wild-type, myrf-1(ybq98); Pdpy-7-Cre(tmIs1028) and myrf-1(ybq98)/mIn1; Pdpy-7-Cre(tmIs1028) animals cultured at 20 °C for 48 hours, starting from freshly laid eggs, as illustrated in *Figure 4D*.

embryogenesis and by distinct yet unidentified mechanisms during early L1, this result suggests that MYRF-1 plays a predominant role in promoting *lin-4* transcription.

In our previous study, we characterized a mutant of *myrf-1*1-700(*syb1313*) that exhibited precocious synaptic remodeling and M-cell division during mid-late L1, albeit in a discordant manner (*Xia et al., 2021*). Our analysis of MYRF-11-700 localization led us to infer that MYRF-1(*syb1313*) undergoes

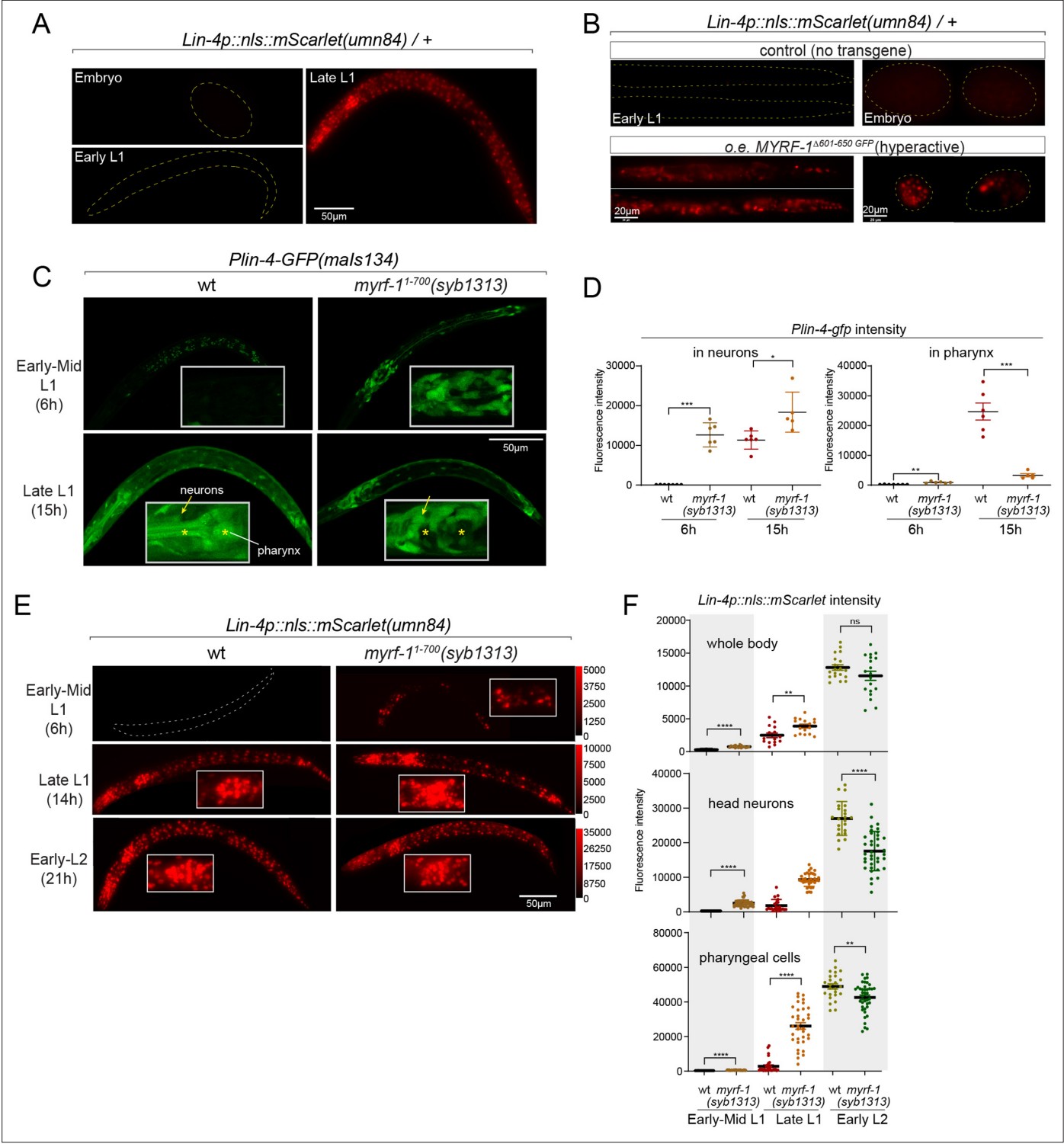

**Figure 5.** Hyperactive MYRF-1 drives premature expression of *lin-4*. (**A**). The reporter of *lin-4* transcription, labeled by endogenously inserted *nls::mScarlet (umn84)*, which also produces a loss-of-function allele of *lin-4*. The fluorescence was not observed in embryos or early L1, but in late L1, confirming the previous reports.(**B**). Overexpression of a hyperactive MYRF-1 mutant, GFP::MYRF-1(delete 601–650) caused premature *lin-4* transcription in embryos and early L1, labeled by *lin-4p::nls::mScarlet*.(**C**). The expression of *Plin-4-GFP*(*maIs134*) in wild-type and *myrf-1(syb1313)* mutants. At 6 hr, *Plin-4-GFP* expression is elevated in the neurons of *myrf-1(syb1313)* mutants but undetectable in wild-type. By late L1 (15 hr), *Plin-4-GFP* is upregulated in multiple tissues in wild-type. Although GFP expression is sustained in neurons (arrow) of the mutants, it is significantly weak or absent in the pharynx (asterisk) of the mutants.(**D**). The fluorescence intensity of the *lin-4* transcriptional reporter (as displayed in (**C**)) was measured and presented as mean ± SEM (t-test, *p<0.05, **p<0.01, ***p<0.001). Each data point represents the mean intensity of the head neurons or pharynx region in individual

*Figure 5 continued on next page*

*Figure 5 continued*

animal, which were imaged using confocal microscopy. (*Figure 5—source data 1*).(**E**). The expression of *lin-4p::nls::mScarlet(umn84)* in wild-type and *myrf-1(syb1313)* mutants. At 6 hr, mScarlet expression is elevated in certain neurons of *myrf-1(syb1313)* mutants but undetectable in wild-type. By late L1 (14 hr), mScarlet is upregulated in multiple tissues in both wild-type and *myrf-1(syb1313)* mutants. The mutants exhibit stronger mScarlet signals than wild-type.(**F**). The fluorescence intensity of *lin-4p::nls::mScarlet* (as displayed in (**E**)) was measured and presented as mean ± SEM (t-test, *p<0.05, **p<0.01, ***p<0.001). The ROI for 'whole body' encompasses the entire body of the animal. The ROI of 'in head neurons' comprises multiple head neuron nuclei in each animal. The region of interest (ROI) for 'in pharyngeal cells' includes the 8–9 pharyngeal nuclei that exhibit strong mScarlet signals. Each data point represents the mean intensity of the ROI in individual animal. (*Figure 5—source data 2*).

The online version of this article includes the following source data for figure 5:

**Source data 1.** The detailed statistical analysis of the fluorescence intensity of the lin-4 transcriptional reporter(maIs134) in wild type and myrf-1(syb1313) mutant, as illustrated in *Figure 5D*.

**Source data 2.** The detailed statistical analysis of the fluorescence intensity of lin-4p::nls::mScarlet in wild-type and myrf-1(syb1313) mutants, as illustrated in *Figure 5F*.

unregulated cleavage processing, resulting in the release of a small quantity of trimer N-MYRF (*Figure 1D and E*). However, this processing is inefficient and inconsistent across tissues. In the current study, we investigated the expression of the *lin-4* transcription reporter (*maIs134*) in *myrf-1(syb1313)* and observed a clear appearance of the signal in the mutants, especially in neurons, at 6 hr, while it was undetectable in wild-type animals at this stage (*Figure 5C and D*). It is worth noting that at the time (15 hr) when *lin-4* is typically upregulated in wild-type animals, the reporter GFP expression is significantly absent in the pharynx and epidermis of *myrf-1(syb1313)* mutants. However, there is sustained, higher-than-wild-type expression of the reporter GFP in neurons of these mutants. Considering the precocious phenotype observed in DD neurons in *myrf-1(syb1313)*, these data support that hyperactive MYRF-1 promotes the premature transcription of *lin-4*.

We conducted further analysis to investigate the impact of *myrf-1$^{1-700}$(syb1313)* on the endogenously tagged mScarlet *lin-4* transcription reporter. In the mutants, we observed premature induction of mScarlet signals in a subset of nuclei, likely neurons based on their position and nucleus size, at 6 hr (*Figure 5E and F*). As mutant animals progressed towards late L1 (14 hr), the signal intensity significantly increased and remained consistently higher compared to the wild-type controls. From this stage on, while the mScarlet signals in many wild-type individuals were still in the process of upregulation, the signal intensity in *myrf-1$^{1-700}$(syb1313)* mutants did not exhibit a comparable sustained increase. This observation aligns with our assessment of *myrf-1$^{1-700}$(syb1313)* mutants, indicating an inconsistent precocity and developmental progression deficiency. Nonetheless, these results strongly support the notion that hyperactive MYRF can precociously activate endogenous *lin-4* transcription.

Notable differences in expression patterns are observed between the *maIs134* and *umn84* reporters in *myrf-1$^{1-700}$(syb1313)*. Contrary to the absence of *maIs134* signals in the pharynx of mutants, mScarlet signals in *umn84* show premature induction in the pharynx at 6 hr (which is absent in wild-type at a similar stage), and exhibit stronger expression at 14 hr compared to wild-type (*Figure 5E and F*). One possible explanation for this contrasting expression pattern is the presence of additional pharynx-enhancing elements located outside the 2.4 kb promoter region of *lin-4* used in the *maIs134* transgene. Alternatively, it should be considered that the threshold for transcription factor activation required to drive the expression of the endogenous reporter (*umn84*) versus the multicopy DNA array reporter (*maIs134*) is likely different. Therefore, resolving this discrepancy would require further investigation.

## MYRF-1 interacts with *lin-4* promoter directly

While analyzing *Plin-4-GFP (maIs134)*, we observed that a subset of the animals carrying the transgene enter dauer even when the food is still available (*Figure 6A and B*; *Figure 6—figure supplement 1*). Dauer refers to the alternative L3 stage that animals develop into when they encounter unfavorable living conditions such as low food abundance and high temperature. In normal laboratory conditions, wild-type *C. elegans* would never become dauer before the food runs out. We also observed that animals carrying *maIs134* invariably develop slower than wild-type animals, even though they show no obvious defect in becoming adults and in fertility (*Figure 6B–E*). The developmental delay starts from L2 and onwards based on their appearance and body length measurement. The L2 animals of *maIs134*

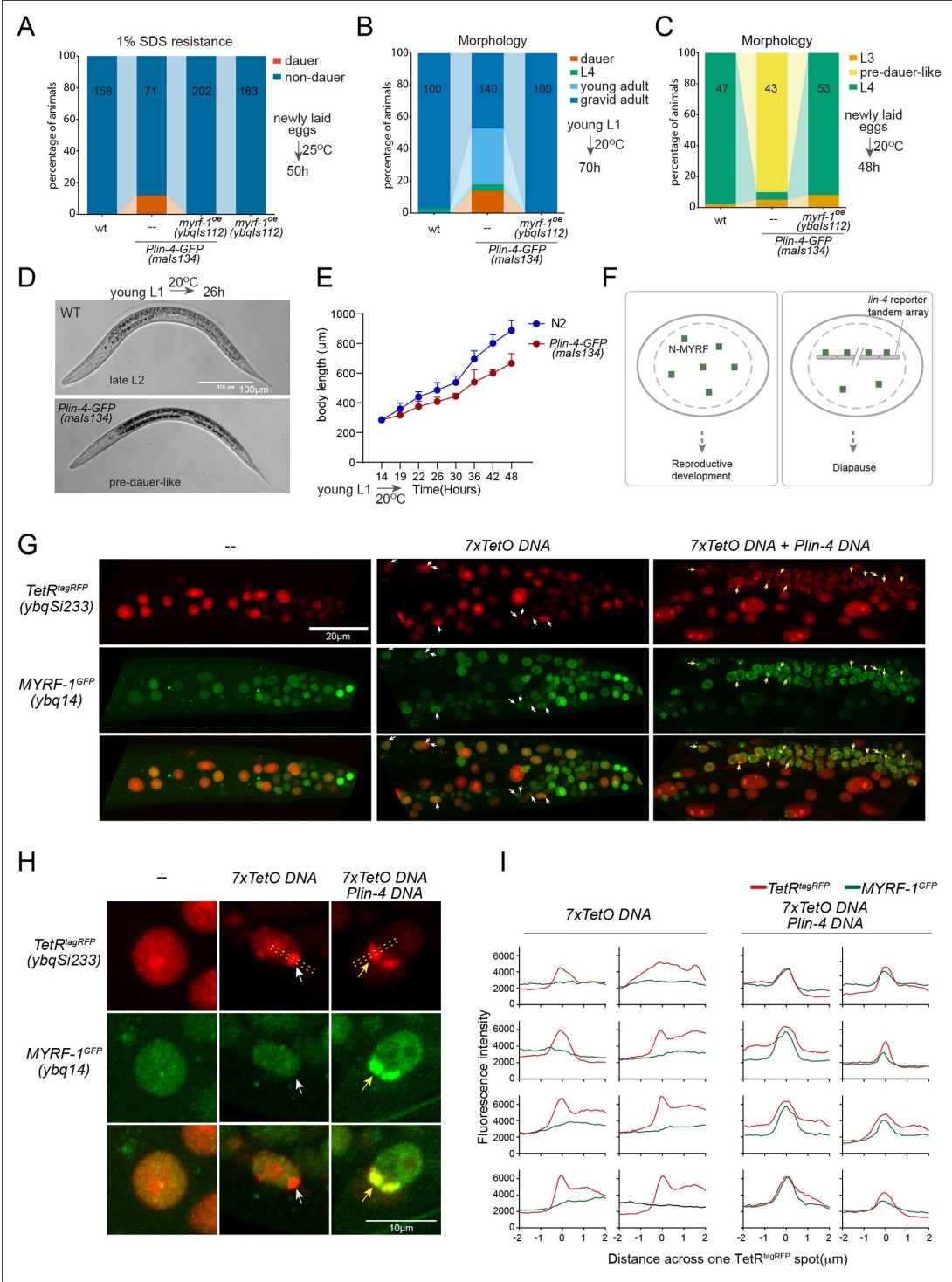

**Figure 6.** *lin-4* promoter DNA recruits MYRF-1 protein in vivo. (**A**) A subset of animals carrying the *maIs134* transgene constitutive dauer formation when food is still available on culture plates. Dauer formation was assessed by treating the animals with 1% SDS for 20 min. MYRF-1 overexpression (*ybqIs112*) suppresses the constitutive dauer formation in *maIs134*. Animals were cultured at 25 °C for 50 hr starting from freshly laid eggs. The number of animals analyzed is indicated on each bar. (**Figure 6—source data 1**). (**B**) Morphological assessment shows that a subpopulation of animals carrying the maIs134 transgene becomes dauer larvae, which exhibit a lean body and darkened intestine (**Figure 3**). MYRF-1 overexpression (*ybqIs112*) suppresses the constitutive dauer formation in *maIs134*. Animals were cultured at 20 °C for 70 hr starting from young L1. The number of animals analyzed is indicated on each bar. (**Figure 6—source data 2**). (**C**) The development of *maIs134* is delayed compared to wild-type animals. with the majority of *maIs134* animals exhibiting pre-dauer-like characteristics while most of the wild-type animals become L4. Animals were cultured at 20 °C for 48 hr starting from freshly laid eggs. (**Figure 6—source data 3**). (**D**) Representative images of animals from experiments in C. At 26 hr *maIs134* animals are thinner than wild-

*Figure 6 continued on next page*

*Figure 6 continued*

type and have dark intestinal granules, which are characteristic of pre-dauer (L2d). (**E**) Measurements of body length of wild-type and *maIs134* animals show a growth delay in *maIs134* starting from L2. The mean body length of analyzed animals at a series of time points is shown on the graph, with the mean ± SD indicated. (***Figure 6—source data 4***). (**F**) A hypothetical model suggests that the tandem DNA array of the *lin-4* promoter sequesters the MYRF protein, resulting in a decreased availability of MYRF to regulate its normal transcription targets. (**G**) A tandem DNA array containing *lin-4* promoter (2.4 kb) DNA causes puncta of GFP::MYRF-1(*ybq14*) in the nucleus. As a control, a 7xTetO sequence-containing DNA array causes the puncta formation of TetR::tagRFP(*ybqSi233*) (white arrows), while it does not cause the aggregation of GFP::MYRF-1. Only the addition of *lin-4* promoter DNA causes the formation of GFP::MYRF-1 puncta (yellow arrows). (**H**) Representative images of animal cells carrying transgenes described in G, but in a high magnification view. (**I**) Line plots of signal intensity measurements along the bar ROI drawn across one red punctum in images, examples of which were shown in H. The bar region of interest (ROI) is centered at the fluorescent spot and examples of bar ROI were shown in H. Each individual panel represents signals from one cell. (***Figure 6—source data 5***).

The online version of this article includes the following source data and figure supplement(s) for figure 6:

**Source data 1.** Measurements of the percentage of 1% SDS-resistant dauers for wild-type, maIs134, myrf-1(ybqIs112), and maIs134;myrf-1(ybqIs112) animals cultured at 25 °C for 50 hours, starting from freshly laid eggs, as illustrated in ***Figure 6A***.

**Source data 2.** Assessments of developmental stages for wild-type, maIs134, and maIs134;myrf-1(ybqIs112) animals at 70 hours post-hatching at 20 °C, as illustrated in ***Figure 6B***.

**Source data 3.** Assessments of developmental stages for wild-type, maIs134, and maIs134;myrf-1(ybqIs112) animals cultured at 20 °C for 48 hours, starting from freshly laid eggs, as illustrated in ***Figure 6C***.

**Source data 4.** Measurements of body length for wild-type and maIs134 animals at various stages, as illustrated in ***Figure 6E***.

**Source data 5.** The detailed statistical analysis of the fluorescence intensity for the GFP::MYRF-1 and tetR::RFP fluorescent spots, as illustrated in ***Figure 6I***.

**Figure supplement 1.** Constitutive dauer phenotype observed in a subpopulation of *Plin-4-GFP(maIs134)* animals.

**Figure supplement 2.** *lin-4* promoter DNA causes aggregation of GFP::MYRF-1(*ybq14*) in the nucleus.

**Figure supplement 2—source data 1.** The detailed statistical analysis of the fluorescence intensity for the bar ROI drawn across the GFP::MYRF-1 spots, as illustrated in ***Figure 6—figure supplement 2C***.

are typically darker, longer, fatter than normal L2 animals and resemble pre-dauer (L2d) animals in gross morphology (***Figure 6D***; ***Figure 6—figure supplement 1***).

Although constitutive dauer formation can be affected by a range of factors, given the crucial role of MYRF in facilitating larval development, we explored the possibility that the dauer formation observed in *maIs134* might be due to the tandem array transgene of the *lin-4* promoter DNA capturing a substantial amount of MYRF-1 protein. This could lead to a significant drop in MYRF-1 levels, hindering normal development. If the hypothesis was true, expressing more MYRF-1 in *maIs134* animals might suppress the dauer phenotype. In the rescuing transgene *myrf-1*^LoxP^(*ybqIs112*) generated in our previous studies (***Meng et al., 2017***), the MYRF-1 protein is observed to increase slightly beyond its endogenous level. We find that *ybqIs112* transgene completely suppresses the slow-growth phenotype in *maIs134*, as well as dauer formation (***Figure 6A–C***). This suggests that the transgene containing *lin-4* promoter DNA negatively interferes with the normal function of MYRF-1 in development (***Figure 6F***).

These observations prompted us to investigate whether the transgene array of *lin-4* promoter DNA binds a significant amount of MYRF-1 protein. To test this, we injected the 2.4 kb *lin-4* promoter RFP reporter plasmid or the reporter without RFP into *myrf-1*^GFP^(*ybq14*), and remarkably, we detected some puncta of intensified GFP signals in a subset of nuclei in animals carrying transgene (***Figure 6G–I***; ***Figure 6—figure supplement 2***). This suggests that the MYRF-1^GFP^ protein is concentrated in discrete nuclear locations, a phenomenon that we never observed in wild-type *myrf-1*^GFP^(*ybq14*) animals. To determine the specificity of their interaction, we designed a second plasmid containing 7 x TetO sequence to be bound by TetR. The TetR DNA binding domain with a RFP tag (TetR::RFP) is driven by a ubiquitous promoter, in a form of single copy transgene (*ybqSi233*) to ensure consistent, moderate expression (***Figure 6G–I***). The formation of the TetO tandem DNA array indeed led to the appearance of TetR::RFP puncta, demonstrating the effectiveness of the method. With such a method, the presence of RFP puncta can then mark the location of DNA arrays in the nucleus. The formation of the TetO DNA array alone was insufficient to sequester MYRF-1^GFP^, while only the addition of the *lin-4* promoter DNA into the tandem array caused the emergence of the MYRF-1^GFP^ puncta, indicating the specificity of the *lin-4* promoter-MYRF-1 interaction. Therefore, the serendipitous observations

concerning Daf-C in *maIs134* and the nuclear loci co-labeling in vivo supports the direct regulator role of MYRF-1 in driving *lin-4* transcription.

## MYRF-1 regulates a selective subset of microRNAs during L1-L2 transition

We wanted to investigate the extent to which MYRF-1's transcriptional activity might affect the landscape of microRNA expression in genome, as microRNAs often have stronger effects when present in combination with homologous microRNAs or in synergy with other microRNA families (*Ambros and Ruvkun, 2018*). We performed microRNA-targeted sequencing analysis on *myrf-1(ju1121)* mutants of late L1 (16 hr) compared to controls. The sequencing analysis showed that a small subset of microRNA species was differentially expressed between *myrf-1(ju1121)* and control animals (*P<0.05*) (*Figure 7A*). We performed a phylogenetic analysis to analyze the relationships between these microRNA species (*Figure 7B*). Notably, 6 of the 7 microRNAs showing increased expression in *myrf-1(ju1121)* compared to wild-type are clustered on a single phylogenetic branch, distinct from the other two branches. Among the differentially expressed, *lin-4* was the most decreased gene, consistent with our present analysis on *lin-4*. We then selected several candidates of relative abundance and examined their expression using transcriptional reporters. We generated single copy insertion transgenes carrying 2 kb upstream sequences of the candidate microRNA and eventually obtained lines for *mir-48*, *mir-73*, and *mir-230* showing consistent GFP signals (*Figure 7C and D*). The reporter for *mir-48* is primarily detected in the pharynx, *mir-73* is present in both the pharynx and seam cells, whereas *mir-230* is detected in seam cells. The *mir-48* reporter (*ybqSi206*) was decreased, while *mir-73* (*ybqSi208*) and *mir-230* (*ybqSi209*) were increased in *myrf-1(ju1121)* at early L2, thus confirming the microRNA sequencing results. Notably, all three microRNAs were increased from L1 to L2 by transcriptional reporter; among the three, *mir-48* reporter is hardly detected in L1 and shows the most dramatic increase when animals transition to L2. The loss of *myrf-1* significantly decreased *mir-48*'s upregulation, but did not completely block it as it did with *lin-4*. *mir-48* belongs to the *let-7* microRNA family, and two other family members, *mir-84* and *mir-241*, were also downregulated in *myrf-1(ju1121)*, while *let-7*, the founding member, itself was upregulated in *myrf-1(ju1121)*. Among the microRNAs upregulated in the *myrf-1* mutants, *mir-42* belongs to the *mir-35* family, which is essential for embryogenesis, and *mir-71* has been extensively studied for its upregulation in adults and regulatory role in aging. At this stage, it is unclear how these candidate microRNAs may be involved in regulating post-embryonic development, but our results suggest an intricate genetic circuit in which MYRF-1's activity may either enhance or dampen the expression of select microRNA species.

## Discussion

Our study reveals that the nuclear accumulation of released N-terminal MYRF-1 increases from mid L1 to late L1, coinciding with the activation of *lin-4* expression. We demonstrated that MYRF-1 is essential for the expression of *lin-4* in terminal tissues in L1 and also required for *lin-4* expression throughout the larval stages (*Figure 7E*). Our data support the direct, predominant role of MYRF-1 in driving *lin-4* transcription.

While *lin-4* expression appears to be constitutive after late L1 by stable GFP reporter, the analysis with higher temporal resolution showed that it exhibits oscillation in each larval stage (*Kim et al., 2013*; *Stec et al., 2021*). The periodic activation of *lin-4* transcription is likely driven by poorly understood oscillation gene regulatory circuits, some of which include *C. elegans* orthologs of human circadian clock genes (*Hendriks et al., 2014*; *McCulloch and Rougvie, 2014*; *Meeuse et al., 2020*; *Perales et al., 2014*; *Tsiairis and Großhans, 2021*; *Van Wynsberghe et al., 2014*). Recent work has begun to reveal the regulation mechanisms underlying oscillating *lin-4* transcription (*Kinney et al., 2023*; *Perales et al., 2014*; *Stec et al., 2021*). BLMP-1/PRDM1 functions as a nutrient-stimulated priming factor to decompact the *lin-4* locus and boost the amplitude of *lin-4* oscillation (*Stec et al., 2021*). It is worth noting that the expression of BLMP-1 itself does not oscillate. However, two other oscillating nuclear hormone receptors, NHR-85/Rev-Erb, and NHR-23/ROR, promote *lin-4* transcription, ensuring its precise oscillation timing and proper dosage (*Kinney et al., 2023*). These two receptors are the closest nematode orthologs of human circadian transcription factors Rev-Erb and ROR, respectively. The expression of NHR-85 is downregulated post-transcriptionally by LIN-42/PERIOD,

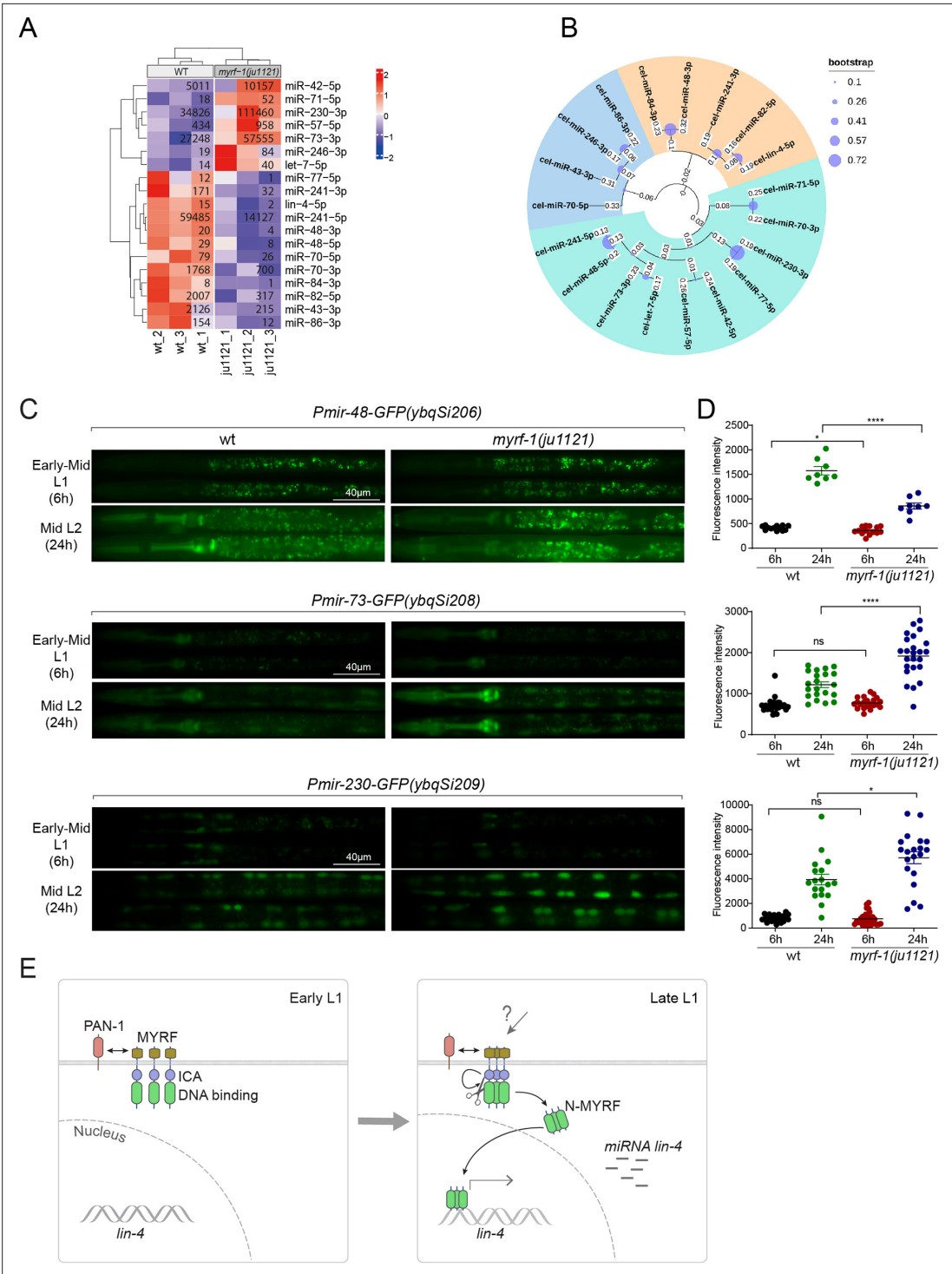

**Figure 7.** MYRF-1 regulates a selective subset of microRNAs during L1-L2 transition. (**A**) The heatmap displays hierarchically clustered differentially expressed miRNAs between wild-type and *myrf-1(ju1121)* animals (p<0.05) at the late L1 stage. It is generated from rlog-transformed TPM (Transcripts Per Million) values of miRNA-seq data, followed by scaling using z-scores per gene (row). In the heatmap, red indicates increased expression, while blue denotes decreased expression. The displayed numbers are the average TPM values for each specific miRNA, calculated from three replicates. (***Figure 7—source data 1***). (**B**) The phylogenetic analysis shows the relationship between differentially expressed miRNA genes in *myrf-1(ju1121)*. The three branches are color-coded.(**C**) The expression of transcriptional reporters, *Pmir-48-gfp(ybqSi206)*, *Pmir-73-gfp(ybqSi208)*, and *Pmir-230-gfp(ybqSi209)* in wild-type and *myrf-1(ju1121)* animals at early-mid L1 (6 hr) and middle L2 (24 hr) is shown. (**D**) The fluorescence intensity of transcriptional reporters shown in (**C**) is quantified and presented as mean ± SEM (t-test: ns, not significant, *p<0.05, ****p<0.0001). (***Figure 7—source data 2***). (**E**) An illustration of how MYRF is processed and promotes *lin-4* expression during development. At early L1, MYRF is mainly localized to the cell membrane in a PAN-1-

*Figure 7 continued on next page*

*Figure 7 continued*

dependent manner. However, during late L1, N-MYRF is released from the membrane through the catalytic activity of the ICE (Intramolecular Chaperone of Endosialidase) domain following its trimerization. Subsequently, N-MYRF translocates to the nucleus to enhance the transcription of *lin-4*.

The online version of this article includes the following source data for figure 7:

**Source data 1.** The source data for microRNA read counts, as illustrated in *Figure 7A*.

**Source data 2.** The detailed statistical analysis of the fluorescence intensity for the mir-48, mir-73, and mir-230 transcriptional reporters is illustrated in *Figure 7D*.

another ortholog of the human circadian factor. LIN-42/PERIOD peaks in each cycle, but lags behind NHR-85 and NHR-23 (*Kinney et al., 2023*). Although the interactions between BLMP-1, NHR-85, and NHR-23 constitute an attractive model for explaining the oscillation of *lin-4* expression, it remains unclear to what extent the activation of *lin-4* expression, particularly during late L1, relies on these factors. For instance, there is a lack of description regarding the impact of BLMP-1, NHR-85, and NHR-23 mutants on endogenous *lin-4* expression. Exploring how MYRF-1 interacts with the three components of *lin-4* oscillation control—BLMP-1, the NHR-85 and NHR-23 pair, and LIN-42—would be an interesting direction for further study.

Similar to the orthologs of circadian genes, *myrf-1* mRNA exhibits oscillation in each larval stage as part of an oscillation scheme consisting of over 3000 genes oscillating at specific phases in each cycle (*Hendriks et al., 2014*; *Meeuse et al., 2020*). These gene expression patterns, including *lin-4*, are halted during acute food removal/starvation, indicating that cycle progression is regulated by nutritional state (*Schindler et al., 2014*; *Stec et al., 2021*). Our observations suggest that the tandem array of *lin-4* promoter DNA may sequester a certain amount of MYRF protein. This sequestration could limit the availability of MYRF, potentially leading to a partial dauer-constitutive phenotype. Although we do not fully understand the phenomenon, it points to a link between MYRF-1 activity and nutrient state. Investigating how MYRF-1 responds to nutrient signals, what transcriptional outcome it produces, and its position in the regulatory circuit of oscillation will be important topics for future investigation.

## Limitations of the study

While the tandem DNA array of the *lin-4* promoter can sequester MYRF-1 proteins in vivo, forming discrete nuclear loci, we have not yet obtained further direct evidence for MYRF-1's binding to the endogenous *lin-4* promoter. Chromatin immunoprecipitation (ChIP) is commonly utilized to determine whether a transcription factor binds to a DNA element in vivo. However, we have not yet obtained conclusive ChIP results, noting that MYRF is present in low quantities and highly susceptible to degradation. These experiments remain part of our ongoing pursuit to identify MYRF's transcription targets.

# Materials and methods

**Key resources table**

| Reagent type (species) or resource | Designation | Source or reference | Identifiers | Additional information |
|---|---|---|---|---|
| Gene (*Caenorhabditis elegans*) | myrf-1 | WormBase Gene ID | WBGene00004134 | other name: pqn-47 |
| Gene (*Caenorhabditis elegans*) | myrf-2 | WormBase Gene ID | WBGene00008999 | |
| Gene (*Caenorhabditis elegans*) | pan-1 | WormBase Gene ID | WBGene00003915 | |
| Gene (*Caenorhabditis elegans*) | lin-4 | WormBase Gene ID | WBGene00002993 | |
| Gene (*Caenorhabditis elegans*) | mir-48 | WormBase Gene ID | WBGene00003039 | |
| Gene (*Caenorhabditis elegans*) | mir-73 | WormBase Gene ID | WBGene00003301 | |

*Continued on next page*

*Continued*

| Reagent type (species) or resource | Designation | Source or reference | Identifiers | Additional information |
|---|---|---|---|---|
| Gene (*Caenorhabditis elegans*) | mir-230 | WormBase Gene ID | WBGene00003323 | |
| Chemical compound, drug | G418 | BBI LIFE SCIENCES | A600958-0001 | 25 mg/mL |
| Commercial assay or kit | ClonExpress II One Step Cloning | Vazyme Biotech Co., Ltd | C112-01 | |
| Commercial assay or kit | FastPure Plasmid Mini Kit | Vazyme Biotech Co., Ltd | DC201-01 | |
| Commercial assay or kit | Taqman microRNA RT kit | ThermoFisher | 4366596 | |
| Commercial assay or kit | TAKARA RNA reverse transcription kit | TAKARA | 6110 A | |
| Strain, strain background (*Caenorhabditis elegans*) | *lin-4*(umn84[lin-4p::SL1::EGL-13NLS::lox2272::mScarlet-I::cMycNLS::Lox511I::let-858 3'UTR::lox2722])/mIn1[dpy-10(e128) umnIs33] II. | *Caenorhabditis* Genetics Center | CGC177 | *Figure 2C–H*, *Figure 2—figure supplement 2*, *Figure 2—figure supplement 3A–D*, *Figure 5A, B, E and F* |
| Strain, strain background (*Caenorhabditis elegans*) | unc-119(ed3) III; lin-4p::GFP +unc-119(+)(maIs134). | *Caenorhabditis* Genetics Center | VT1072 | *Figure 2A–B*, *Figure 4A*, *Figure 5C–D*, *Figure 6A–D*, *Figure 2—figure supplement 1*, *Figure 6—figure supplement 1* |
| Strain, strain background (*Caenorhabditis elegans*) | lin-14::GFP(cc2841) X. | *Caenorhabditis* Genetics Center | PD1301 | *Figure 3A–B* |
| Strain, strain background (*Caenorhabditis elegans*) | GFP::pqn-47::3xFlag(ybq14) II; glo-1(zu391). | This paper; The Yingchuan B. Qi Laboratory | BLW1827 | *Figure 1A–B*, *Figure 6—figure supplement 2A–C* |
| Strain, strain background (*Caenorhabditis elegans*) | myrf-1(ju1121) / mIn1 II; lin-4p::GFP +unc-119(+)(maIs134). | This paper; The Yingchuan B. Qi Laboratory | BLW1424 | *Figure 2A–B*, *Figure 4A* |
| Strain, strain background (*Caenorhabditis elegans*) | myrf-1(ju1121) II / [mIs14 dpy-10(e128)](mIn1) II; lin-14::GFP(cc2841) X. | This paper; The Yingchuan B. Qi Laboratory | BLW1410 | *Figure 3A–B* |
| Strain, strain background (*Caenorhabditis elegans*) | *lin-4*(umn84[lin-4p::SL1::EGL-13NLS::lox2272::mScarlet-I::cMycNLS::Lox511I::let-858 3'UTR::lox2722]) II myrf-1(ybq6) II /mIn1[dpy-10(e128) umnIs33] II. | This paper; The Yingchuan B. Qi Laboratory | BLW2284 | *Figure 2C–D*, *Figure 2—figure supplement 3A* |
| Strain, strain background (*Caenorhabditis elegans*) | *lin-4*(umn84[lin-4p::SL1::EGL-13NLS::lox2272::mScarlet-I::cMycNLS::Lox511I::let-858 3'UTR::lox2722]) II myrf-1(ybq6) II /mIn1[dpy-10(e128) umnIs33] II; myrf-2(ybq42) X. | This paper; The Yingchuan B. Qi Laboratory | BW2285 | *Figure 2C–D*, *Figure 2—figure supplement 3A* |
| Strain, strain background (*Caenorhabditis elegans*) | *lin-4* umn84[lin-4p::SL1::EGL-13NLS::lox2272::mScarlet-I::cMycNLS::Lox511I::let-858 3'UTR::lox2722 / dpy-10(e128) (mT1) II, III; pan-1(gk142) III / dpy-10(e128), Pmyo-2-mCherry_chr_2(mT1-mCherry#2) III. | This paper; The Yingchuan B. Qi Laboratory | BLW2300 | *Figure 2G–H*, *Figure 2—figure supplement 3B* |

*Continued on next page*

*Continued*

| Reagent type (species) or resource | Designation | Source or reference | Identifiers | Additional information |
|---|---|---|---|---|
| Strain, strain background (*Caenorhabditis elegans*) | *lin-4*(umn84[lin-4p::SL1::EGL-13NLS::lox2272::mScarlet-I::cMycNLS::Lox511I::let-858 3'UTR::lox2722]) II /mln1[dpy-10(e128) umnIs33] II; myrf-2(ybq42) X. | This paper; The Yingchuan B. Qi Laboratory | BLW2385 | *Figure 2C–D*, *Figure 2—figure supplement 3A* |
| Strain, strain background (*Caenorhabditis elegans*) | *lin-4*(umn84[lin-4p::SL1::EGL-13NLS::lox2272::mScarlet-I::cMycNLS::Lox511I::let-858 3'UTR::lox2722]) II myrf-1(ju1121) II /mln1[dpy-10(e128) umnIs33] II. | This paper; The Yingchuan B. Qi Laboratory | BLW2236 | *Figure 2E–F*, *Figure 2—figure supplement 3C–D* |
| Strain, strain background (*Caenorhabditis elegans*) | *lin-4*(umn84[lin-4p::SL1::EGL-13NLS::lox2272::mScarlet-I::cMycNLS::Lox511I::let-858 3'UTR::lox2722]) II myrf-1(ybq1313) II /mln1[dpy-10(e128) umnIs33] II. | This paper; The Yingchuan B. Qi Laboratory | BLW2237 | *Figure 5E–F* |
| Strain, strain background (*Caenorhabditis elegans*) | myrf-1(ju1121) II / [mIs14 dpy-10(e128)](mln1) II; lin-4p::GFP +unc-119(+)(maIs134; Pmyo-3-MYRF-1(ybqEx746). | This paper; The Yingchuan B. Qi Laboratory | BLW1579 | *Figure 4A* |
| Strain, strain background (*Caenorhabditis elegans*) | myrf-1(ju1121) II / [mIs14 dpy-10(e128)](mln1) II; lin-4p::GFP +unc-119(+)(maIs134; Pdpy-7-GFP::MYRF-1(ybqEx721). | This paper; The Yingchuan B. Qi Laboratory | BLW1580 | *Figure 4A* |
| Strain, strain background (*Caenorhabditis elegans*) | gfp::myrf-1(LoxP) (ybq98) II. | This paper; The Yingchuan B. Qi Laboratory | BLW1934 | *Figure 4B–E* |
| Strain, strain background (*Caenorhabditis elegans*) | gfp::myrf-1(LoxP)(ybq98) / [mIs14 dpy-10(e128)](mln1) II; Pdpy-7-NLS::Cre(tmIs1028); lin-4p::GFP +unc-119(+)(maIs134). | This paper; The Yingchuan B. Qi Laboratory | BLW2020 | *Figure 4B–E* |
| Strain, strain background (*Caenorhabditis elegans*) | GFP::MYRF-1 (1–700)::3xFlag(syb1313) II / [mIs14 dpy-10(e128)](mln1) II; lin-4p::GFP +unc-119(+)(maIs134). | This paper; The Yingchuan B. Qi Laboratory | BLW2037 | *Figure 5C–D* |
| Strain, strain background (*Caenorhabditis elegans*) | Ppqn-47-LoxP-NLS::tagRFP::T2A::pqn-47-LoxP(ybqIs112) V. | This paper; The Yingchuan B. Qi Laboratory | BLW1235 | *Figure 6A* |
| Strain, strain background (*Caenorhabditis elegans*) | lin-4p::GFP +unc-119(+)(maIs134); Ppqn-47-LoxP-NLS::tagRFP::T2A::pqn-47-LoxP(ybqIs112) V. | This paper; The Yingchuan B. Qi Laboratory | BLW2172 | *Figure 6A–C*, *Figure 6—figure supplement 1A–B* |
| Strain, strain background (*Caenorhabditis elegans*) | GFP::pqn-47::3xFlag(ybq14) II, glo-1(zu391), Prpl-28-tetR-tagRFP (ybqSi233). | This paper; The Yingchuan B. Qi Laboratory | BLW2170 | *Figure 6G–I* |
| Strain, strain background (*Caenorhabditis elegans*) | Prpl-28-tetR-tagRFP (ybqSi233). | This paper; The Yingchuan B. Qi Laboratory | BLW2168 | Cross with BLW1827 |
| Strain, strain background (*Caenorhabditis elegans*) | Pmir-48-GFP(ybqSi206). | This paper; The Yingchuan B. Qi Laboratory | BLW2113 | *Figure 7C–D* |
| Strain, strain background (*Caenorhabditis elegans*) | Pmir-73-GFP(ybqSi208). | This paper; The Yingchuan B. Qi Laboratory | BLW2115 | *Figure 7C–D* |

*Continued on next page*

*Continued*

| Reagent type (species) or resource | Designation | Source or reference | Identifiers | Additional information |
|---|---|---|---|---|
| Strain, strain background (*Caenorhabditis elegans*) | Pmir-230-GFP(ybqSi209). | This paper; The Yingchuan B. Qi Laboratory | BLW2116 | *Figure 7C–D* |
| Strain, strain background (*Caenorhabditis elegans*) | myrf-1(ju1121) II / [mIs14 dpy-10(e128)](mIn1) II, Pmir-48-GFP(ybqSi206). | This paper; The Yingchuan B. Qi Laboratory | BLW2181 | *Figure 7C–D* |
| Strain, strain background (*Caenorhabditis elegans*) | myrf-1(ju1121) II / [mIs14 dpy-10(e128)](mIn1) II, Pmir-73-GFP(ybqSi208). | This paper; The Yingchuan B. Qi Laboratory | BLW2182 | *Figure 7C–D* |
| Strain, strain background (*Caenorhabditis elegans*) | myrf-1(ju1121) II / [mIs14 dpy-10(e128)](mIn1) II, Pmir-230-GFP(ybqSi209). | This paper; The Yingchuan B. Qi Laboratory | BLW2183 | *Figure 7C–D* |
| Strain, strain background (*Caenorhabditis elegans*) | gfp::myrf-1 (1–656)(syb1468) II / [mIs14 dpy-10(e128)](mIn1) II; lin-4p::GFP +unc-119(+)(maIs134) | This paper; The Yingchuan B. Qi Laboratory | BLW2184 | *Figure 2—figure supplement 1A–B* |
| Strain, strain background (*Caenorhabditis elegans*) | gfp::myrf-1 (1–482)(syb1491) II / [mIs14 dpy-10(e128)](mIn1) II; lin-4p::GFP +unc-119(+)(maIs134) | This paper; The Yingchuan B. Qi Laboratory | BLW2035 | *Figure 2—figure supplement 1A–B* |
| Strain, strain background (*Caenorhabditis elegans*) | gfp::degron::myrf-1::3xflag(ybq133) II; eft-3p::TIR::F2 A:mTagBFP2::AID*::NLS::tbb-2 3'UTR(wrdSi23) I; lin-4p::GFP +unc-119(+)(maIs134) | This paper; The Yingchuan B. Qi Laboratory | BLW2157 | *Figure 2—figure supplement 4A–B* |
| Recombinant DNA reagent | Pmyo-3-MYRF-1 | This paper; The Yingchuan B. Qi Laboratory | pQA1094 | *Figure 4A* |
| Recombinant DNA reagent | Pdpy-7-MYRF | This paper; The Yingchuan B. Qi Laboratory | pQA1511 | *Figure 4A* |
| Recombinant DNA reagent | Peft-3-Cas9 U6-myrf-1 sgRNA (for cut MYRF-1 211 K) | This paper; The Yingchuan B. Qi Laboratory | pQA1685 | sgRNA plasmid for constructing BLW1934 [gfp::myrf-1(LoxP) (ybq98)] |
| Recombinant DNA reagent | Peft-3-Cas9 U6-myrf-1 sgRNA (for cut MYRF-1 201 V) | This paper; The Yingchuan B. Qi Laboratory | pQA1686 | sgRNA plasmid for constructing BLW1934 [gfp::myrf-1(LoxP) (ybq98)] |
| Recombinant DNA reagent | myrf-1 LoxP knock-in repair template in pCR8 | This paper; The Yingchuan B. Qi Laboratory | pQA1688 | repair template plasmid for constructing BLW1934 [gfp::myrf-1(LoxP) (ybq98)] |
| Recombinant DNA reagent | Prpl-28-GFP-myrf-1 (1–931 delete 601–650 Ce) | This paper; The Yingchuan B. Qi Laboratory | pQA1922 | *Figure 5B* |
| Recombinant DNA reagent | Prpl-28-tetR-tagRFP in_miniMos | This paper; The Yingchuan B. Qi Laboratory | pQA1896 | plasmid for constructing BLW2168 [Prpl-28-tetR-tagRFP (ybqSi233)] |

*Continued on next page*

*Continued*

| Reagent type (species) or resource | Designation | Source or reference | Identifiers | Additional information |
|---|---|---|---|---|
| Recombinant DNA reagent | 7x-TetO | This paper; The Yingchuan B. Qi Laboratory | pQA1961 | *Figure 6G–I* |
| Recombinant DNA reagent | Plin-4(2412 bp) -unc-54–3'UTR-7x-TetO | This paper; The Yingchuan B. Qi Laboratory | pQA1960 | *Figure 6G–I* |
| Recombinant DNA reagent | Plin-4(2412 bp) -unc-54–3'UTR | This paper; The Yingchuan B. Qi Laboratory | pQA1881 | *Figure 6—figure supplement 2A–C* |
| Recombinant DNA reagent | Pmir-48(2000 bp)-GFP in_miniMos | This paper; The Yingchuan B. Qi Laboratory | pQA1861 | plasmid for constructing BLW2114 |
| Recombinant DNA reagent | Pmir-73(2000 bp)-GFP in_miniMos | This paper; The Yingchuan B. Qi Laboratory | pQA1863 | plasmid for constructing BLW2115 |
| Recombinant DNA reagent | Pmir-230(1919 bp)-GFP in_miniMos | This paper; The Yingchuan B. Qi Laboratory | pQA1864 | plasmid for constructing BLW2116 |
| Recombinant DNA reagent | pGH8 Prab-3::mCherry | The Erik Jorgensen Laboratory | pGH8 | Co-marker plasmid for constructing single-copy integrated strain |
| Recombinant DNA reagent | pCFJ90 Pmyo-2::mCherry | The Erik Jorgensen Laboratory | pCFJ90 | Co-marker plasmid, *Figure 6—figure supplement 2A–C* |
| Recombinant DNA reagent | pCFJ601 Peft-3 Mos1 transposase | The Erik Jorgensen Laboratory | pCFJ601 | Co-marker plasmid for constructing single-copy integrated strain |
| Recombinant DNA reagent | PDD162 Peft-3-Cas9 & Empty sgRNA | The Bob Goldstein Laboratory | PDD162 | sgRNA plasmid template |

## Animals

Wild-type *C. elegans* were Bristol N2 strain. Strains were cultured on NGM plates using standard procedures (*Brenner, 1974*). Unless noted, animals were cultured at 20 °C for assays requiring specific developmental stages. Animals analyzed in this paper were hermaphrodites.

## Naming of the alleles

All alleles generated in Y.B.Q. lab are designated as 'ybq' alleles, and all strains, as 'BLW' strains. 'Ex' denotes transgene alleles of exchromosomal array. 'Is' denotes integrated transgene. 'Si' denotes single-copy integrated transgene. 'syb' alleles (in 'PHX' strains) are generated by genomic editing using CRISPR-Cas9 technique. 'syb' alleles were designed by Y.B.Q. and produced by SunyBiotech (Fuzhou, China).

## myrf-1 alleles by CRISPR-Cas9 editing

For the following described alleles generated by CRISPR-Cas9 editing, Cas9 and gRNA were expressed from plasmids. The positive clones were identified using PCR screening to test singled F1 resulted from microinjection. GFP::myrf-1(LoxP) (ybq98) alleles was generated in the background of strain BLW889 [GFP::myrf-1::3xFlag(ybq14)]. Two artificial introns are inserted into the third exon of myrf-1 gene and each intron carries one LoxP site.

 gRNA target: sgRNA1: TCAAGTCGGCTTCTCTTACGTGG

 sgRNA2: TACGTGGCATCTCCAAAACAGGG

 ybq14(background): …GGAATGCCAAGCCCTGTTTTGGAGATGCCACGTAAG - (insertion point) -AGAAGCCGACTTGACACCCCGTGTGAAACGCCAAGAATCGCTCCAAGCTTTGCTGGTATTG

ACGGATTTCCAGATGAGAATTACAGTCAGCAACAGGCAATCAG - (insertion point) - ATTC TCAA AGTT TCAA GAAG AACA GTGG AGTC CACT GTAT GACA TCAA CGCT CAAC CGCT AC AACAACTTCAA...

ybq98: …GGAATGCCAAGCCCTGT<u>C</u>TT<u>A</u>GA<u>A</u>ATGCCACG<u>C</u>AAG - (gtatgtttcgaatgatactaataATAACTTCGTATAGCATACATTATACGAAGTTATaacataacatagaacattttc ag) – AGAAGCCGACTTGACACCCCGTGTGAAACGCCAAGAATCGCTCCAAGCTTTGCTGGTATTG ACGGATTTCCAGATGAGAATTACAGTCAGCAACAGGCAATCAG - (gtaagtttaaactttctcatactaataATAACTTCGTATAGCATACATTATACGAAGTTATattaactaacgcgctcta tttaaattttcag) – ATTCTCAAAGTTTCAAGAAGAACAGTGGAGTCCACTATACGATATTAACGCTCAACCGCTACA ACAACTTCAA...

GFP::Degron::myrf-1::3xFlag(ybq133) was generated in the background of BLW889 [GFP::myrf-1::3xFlag(ybq14)]. Degron is inserted after the last amino acid of GFP (Lys).

sgRNA: CAATCAACCTACAAACACCCTGG

ybq14(background): …GGGATTACACATGGCATGGATGAACTATACAAA - (insertion point) – GCAGTCAATCAACCTACAAACACCC<u>TGG</u>CTCAACTCAA...

ybq133: …GGGATTACACATGGCATGGATGAACTATACAAA - DEGRON GCAGTCAATCAACCTACAAACACCC<u>TTG</u>CTCAACTCAA...

## Generation of transgene alleles

Pmyo-3-MYRF-1 transgene: The vector pQA1094 [Pmyo-3-myrf-1] was injected to BLW1424 [myrf-1(ju1121)/mIn1 II; lin-4p::GFP +unc-119(maIs134)] at concentration of 0.5 ng/μl. The resulting strain is BLW1579 [myrf-1(ju1121)/mIn1 II; lin-4p::GFP +unc-119(maIs134); Pmyo-3-myrf-1(ybqEx746)]

Pdpy-7-MYRF-1 transgene: The vector pQA1511 [Pdpy-7-gfp::myrf-1] injected to N2 at concentration of 0.5 ng/μl to generate BLW1562 [Pdpy-7-gfp::myrf-1(ybqEx721)]. ybqEx721 was crossed into VT1072 [unc-119(ed3) III; lin-4p::GFP +unc-119(maIs134)] to generate BLW1578 [lin-4p::GFP +unc-119(maIs134); Pdpy-7-gfp::myrf-1(ybqEx721)]. BLW1578 was crossed with BLW1424 [myrf-1(ju1121)/mIn1 II; lin-4p::GFP +unc-119(maIs134)] to generate BLW1580 [myrf-1(ju1121)/mIn1 II; lin-4p::GFP +unc-119(maIs134); Pdpy-7-gfp::myrf-1(ybqEx721)].

Overexpress GFP::MYRF-1 (delete 601–650): The vector pQA1922[Prpl-28-GFP::myrf-1(delete 601–650)] were injected into CGC177 lin-4 umn84[lin-4p::SL1::EGL-13NLS::lox2272::mScarlet-I::cMycNLS::Lox511I::let-858–3'UTR::lox2722] /mIn1[dpy-10(e128) umnIs33] at 10 ng/μl.

Tandem DNA array of Plin-4(2412 bp) DNA: The vector pQA1881 [Plin-4(2412 bp) -unc-54–3'UTR] was injected into BLW1827 [gfp::myrf-1](ybq14); glo-1(zu391)] at 50 ng/μl.

Tandem DNA array of 7xTetO: The vector pQA1961 [7xTetO] was injected into BLW2170 [gfp::myrf-1(ybq14)]; glo-1(zu391); [Prpl-28-TetR-DBD::TagRFP(ybqSi233)] at 50 ng/μl.

Tandem DNA array of Plin-4 DNA +7xTetO: The vector pQA1960 [Plin-4(2412 bp) -unc-54–3'UTR - 7xTetO] were injected into BLW2170 [gfp::myrf-1(ybq14)]; glo-1(zu391); [Prpl-28-TetR-DBD::TagRFP(ybqSi233)] at 50 ng/μl.

Pmir-48-gfp single copy transgene: The pQA1861 [Pmir-48-GFP miniMos_vector] at 50 ng/μl and standard components of miniMos system (see below) was injected into N2 animals to make single copy transgene Pmir-48-GFP(ybqSi206).

Pmir-73-gfp single copy transgene: The pQA1863 [Pmir-73-GFP miniMos_vector] at 50 ng/μl and standard components of miniMos system (see below) was injected into N2 animals to make single copy transgene Pmir-73-GFP(ybqSi208).

Pmir-230-GFP single copy transgene: The pQA1864 [Pmir-230-GFP miniMos_vector] at 50 ng/μl and standard components of miniMos system (see below) was injected into N2 animals to make single copy transgene Pmir-230-GFP(ybqSi209).

TetR-TagRFP single copy transgene: The pQA1896 (Prpl-28-TetR-DBD::TagRFP miniMos_vector) at 50 ng/μl and standard components of miniMos system (see below) was injected into N2 animals to make single copy transgene Prpl-28-TetR::TagRFP (ybqSi233).

## Single-copy integrated transgene allele

The procedure for generating single-copy integrated transgenes using miniMos technology in this study followed the protocol established by *Frøkjær-Jensen et al., 2014*. The injection mixture

contained 10 ng/µl pGH8 [Prab-3::mCherry], 2.5 ng/µl pCFJ90 [Pmyo-2::mCherry], 50 ng/µl pCFJ601 [Peft-3-Mos1-transposase], and 50 ng/µl miniMos plasmid harboring the target gene. The mixture was injected into the gonads of N2 animals. After injection, the nematodes were transferred onto NGM medium and incubated at 25 °C for approximately 48 hr. Subsequently, 500 µl of 25 mg/mL G418 solution was added to each plate to screen for nematodes carrying the target transgene. After 7–10 days, healthy nematodes without mCherry co-markers were selected from the medium where all the food was consumed. Candidate single-copy integrators were grown on G418-containing plates and analyzed for target protein expression. Homozygous nematodes carrying a single-copy transgene were identified by their 100% resistance to G418 toxicity and expression of the target protein.

## TaqMan real-time PCR analysis for miRNA *lin-4*

Samples for Taqman Real-Time PCR Assays were prepared from wild-type N2 and BLW252 [myrf-1(ju1121)/mIn1] strains at three stages: Early L1 (0 h), Late L1 (16 h), and Early L2 (21 h). The worms were screened for size and fluorescence marker of the desired transgene using Biosorter. myrf-1(ju1121) mutants lack the pharyngeal GFP, which serves as a transgene marker of balancer mIn1, as a criterion in sorting process. More than 2000 worms were collected for each sample after the Biosorter process.

RNA extraction was carried out using RNAiso Plus (TaKaRa, Dalian, China) following the manufacturer's instructions with some modifications. Briefly, 1 ml of RNAiso Plus was added to each sample, which was homogenized using a pre-cooled grinder. The homogenization buffer was then transferred to a 1.5 ml microcentrifuge tube and allowed to stand at room temperature (15–30 °C) for 5–10 min. The supernatant was discarded, and 200 µl of chloroform was added and shaken for 30 s until the mixture turned milky. The mixture was then centrifuged at 12,000 g at 4 °C for 15 min, and the supernatant was transferred to a new microcentrifuge tube. To the supernatant, 0.5 ml of isoamylol was added and left to stand for 10 min at room temperature, then centrifuged at 12,000 g at 4 °C for 60 min. After discarding the supernatant, the pellet was washed with 75% ethanol at 12,000 g, 4 °C for 60 min, and then dissolved in 20 µl of DEPC-treated water. The concentration of miRNA was determined using Nanodrop, and its quality was assessed through electrophoresis. Each experiment was performed using at least three replicates.

The TAKARA RNA reverse transcription kit (6110 A) was used to perform RNA reverse transcription following the manufacturer's instructions. The concentration of the resulting product was measured using Nanodrop, and its quality was evaluated through electrophoresis.

The Taqman real-time PCR was conducted according to the TaqMan small RNA assay protocol. Initially, the concentration of each sample was diluted to below 10 ng/µl, followed by reverse transcription PCR using the TaqMan microRNA RT kit (4366596). Subsequently, the Taqman real-time PCR mixture was prepared using Taqman small RNA assay (20 X) with lin-4a (21nt) (ID: 000258) and sn2323 (ID: 001760) as the internal control. Each sample was tested using at least three technical replicates and three biological replicates. The TaqMan real-time PCR was carried out using Applied Biosystems QuantStudio 7 Flex Real-Time PCR System (Thermo Fisher), and all the procedures were followed as per the instructions. To determine the relative changes of each sample in each assay, the *lin-4* Ct of technical replicates was averaged and then subtracted by the average sn2323 Ct to obtain the △CT value for each sample. Within each biological replicate and assay, the difference between the △CT of each sample and the △CT of wild-type at Early L1 was calculated to obtain △△CT. The 2 to the power of △△CT was then calculated to obtain the fold change in each sample compared to the wild-type at Early L1. The fold changes were plotted in *Figure 1*.

## miRNA sequencing and bioinformatics analysis

BLW53 [Phlh-8-GFP(ayIs6)] and BLW1555 [myrf-1(ju1121)/ mIn1; Phlh-8-GFP(ayIs6)] strains were utilized for sample preparation. To isolate myrf-1 mutants, late L1 stage worms were selected based on size and lack of pharyngeal GFP (a balancer mIn1 marker) using Biosorter. Over 10,000 worms were collected for each sample for further experimentation. Three biological replicates for myrf-1 mutants and control animals were included. Total RNA was extracted and assessed for quality and quantity using Nanodrop and Agilent 2100, respectively. Samples with a total quantity of 3–5 µg and high quality (RIN >10) were selected for miRNA library construction. The miRNA libraries were established and sequenced by Novogene Technology Co., Ltd. (Beijing, China).

In this study, the *C. elegans* Assembly WBcel235/ce11 was used as the reference genome for bioinformatic analysis of the sequencing data. The reference genome and gene annotation database were obtained from the Ensembl database and used for downstream analysis. The sequencing data was initially assessed for quality using the software FASTQ, followed by alignment of each miRNA sequencing dataset independently to the reference genome using HiSAT2. Transcriptome from each RNA-sequencing dataset was then extracted using the StringTie software. Differential miRNA cluster analysis was performed to compare miRNA expression of control and myrf-1(ju1121) mutant using DESeq2. Data normalization was done using the TPM method. A P-value was assigned to each gene and adjusted using the Benjamini and Hochberg method. Genes with P_ adj <0.05 were considered differentially expressed.

For the analysis and visualization of miRNA expression, read counts (TPM values) were first normalized to account for library size. Subsequently, variance stabilization was performed using the rlog function from DESeq2 (*Love et al., 2014*). The data were then clustered based on a Euclidean distance metric.

## Microscopic analyses and quantification

Live animals were anesthetized using 0.5% 1-Phenoxy-2-propanol in ddH$_2$O and mounted on 3% Agar gel pad. The animals were examined under 20x, 40x, or 60x oil objective on OLYMPUS BX63 microscope. The wide-field DIC or fluorescence images (single plane images or Z-focal stacks) were taken by a sCMOS camera Prim$\Sigma$ Photometrics camera (model 2) mounted on OLYMPUS BX63, which is driven by CellSens Dimension software (Olympus). Images of live animals were also acquired on Zeiss LSM880 with Airyscan. The thickness of the optical slices was typically 0.8 µm.

For quantification of GFP::MYRF-1 subcellular localization, we conducted three independent rounds of culture, imaging, and scoring. The data were then pooled and presented in percentage column graphs. Regarding quantification of *lin-4* transcriptional reporter intensity, we conducted at least two independent rounds of culture and imaging analysis.

To quantify the patterns of GFP::MYRF-1, we acquired images of stage-synchronized animals using wide-field microscopy as described above. The same parameters were used, including the power of excitation light, an identical objective, exposure duration, and screen display setting. The acquired images were examined, and the patterns of GFP::MYRF were categorized based on the consistency of the signals observed at specific subcellular locations throughout the animal body. A 'weak or unclear signal' means that either no clear signals are detected or there are some weak signals that are inconsistent throughout the animal body.

To quantify the general Plin-4-GFP(maIs134) transcriptional reporter fluorescence intensity, we opened each acquired image stack in ImageJ and selected a single slice with a focused pharynx for further analysis. A square with sides 20 pixels (3.25 µm) long was drawn to the pharynx isthmus. The intensity of the region of interest (ROI) was measured and presented as Mean ± SEM (t-test, *p<0.05; **p<0.01; ***p<0.001; ****p<0.0001).

To quantify the pharyngeal Plin-4-GFP(maIs134) transcriptional reporter fluorescence intensity in myrf-1(syb1491) and myrf-1(syb1468) mutants, we opened each acquired image stack in ImageJ and selected a single slice with a focused pharynx bulb for further analysis. The posterior pharynx bulb was selected as the ROI. The intensity of the ROI was measured and presented as Mean ± SEM (t-test, *p<0.05; **p<0.01; ***p<0.001; ****p<0.0001).

To quantify the neuronal Plin-4-GFP(maIs134) transcriptional reporter fluorescence intensity in myrf-1(syb1491) and myrf-1(syb1468) mutants, we opened each acquired image stack in ImageJ and selected a single slice with a focused head nerve ring for further analysis. The dorsal nerve ring area was selected as the ROI. The intensity of the ROI was measured and presented as Mean ± SEM (t-test, *p<0.05; **p<0.01; ***p<0.001; ****p<0.0001).

The fluorescence intensity of the lin-4p::nls::mScarlet(umn84) transcriptional reporter across the entire body in pan-1(gk142), myrf-1(syb1313), myrf-1(ju1121), myrf-1(ybq6), myrf-2(ybq42), and myrf-1(ybq6); myrf-2(ybq42) double mutants was quantified by maximally projecting the z-stack into a single image using ImageJ. The ROI was defined as the whole animal area. The measured intensity of this ROI is expressed as Mean ± SEM. Statistical significance was determined using a t-test (*p<0.05; **p<0.01; ***p<0.001; ****p<0.0001).

To quantify the pharyngeal lin-4p::nls::mScarlet(umn84) transcriptional reporter fluorescence intensity in myrf-1(ju1121) and myrf-1(syb1313) mutants, we opened each acquired image stack in ImageJ and selected a single slice with a focused pharynx bulb for further analysis. Two posterior pharynx bulb nuclei were selected as the ROI. The intensity of the ROI was measured and presented as Mean ± SEM (t-test, *p<0.05; **p<0.01; ***p<0.001; ****p<0.0001).

To quantify the neuronal lin-4p::nls::mScarlet(umn84) transcriptional reporter fluorescence intensity in myrf-1(ju1121) and myrf-1(syb1313) mutants, we opened each acquired image stack in ImageJ and selected a single slice with a focused nerve ring for further analysis. Several neuron nuclei were selected as the ROI. The intensity of the ROI was measured and presented as Mean ± SEM (t-test, *p<0.05; **p<0.01; ***p<0.001; ****p<0.0001).

To quantify the LIN-14::GFP fluorescence intensity, we opened each acquired image stack in ImageJ and selected a single slice with a focused head neurons for further analysis. The head region was selected as the ROI. The intensity of the ROI was measured and presented as Mean ± SEM (t-test, *p<0.05; **p<0.01; ***p<0.001; ****p<0.0001).

To generate the line plot of TetR::TagRFP and GFP::MYRF-1, we opened each acquired image stack from Zeiss Airyscan in ImageJ and selected a single slice with a focused TetR::TagRFP spot for further analysis. We drew a 2-pixel (~0.1 µm)-thick, 12 µm-long ROI line to cross the center of one bright TetR::TagRFP spot. The middle point of the ROI line was positioned at the center of the spot. We ran the 'plot profile' program located in the Analyze menu and recorded plot data in the RFP and GFP channels, respectively. The final graph was generated in GraphPad Prism 8. Each plot line represented the intensity distribution across a single TetR::TagRFP spot from an independent cell.

To generate a multiple line plot of GFP::MYRF-1, each image stack acquired from Zeiss Airyscan was opened in ImageJ, and a single slice with a focused GFP::MYRF-1 spot was selected. When no obviously bright GFP::MYRF-1 spot was found in the set of image slices, we chose a slice with clearly focused GFP::MYRF-1 signals in nucleus. A 2-pixel (~0.1 µm)-thick, 10µm-long ROI line was drawn to cross the center of the GFP::MYRF-1 punctum (or brightest nucleus signal as we could fine). We then ran the program 'plot profile,'located in the Analyze menu, to record the plot data in the GFP channel. The final graph was generated in GraphPad Prism 8. The maximum intensity position in the ROI line was defined as the '0' position in the X axis. Measurement points within two microns centered at the '0' position were graphed. Each plot line represented the intensity distribution across a single GFP::MYRF-1 spot from an independent cell.

To generate a line plot of epidermal Plin-4-gfp in epidermal knock-out MYRF-1 animals, each acquired image stack was opened in ImageJ and a single slice with a focused seam cell was selected for further analysis. A 2-pixel (~0.1 µm) thick, 12µm-long ROI line was drawn to cross a seam cell in a ventral-to-dorsal direction. The middle point of the ROI line was positioned at the center of the seam cell. The data was recorded by the 'plot profile' program located in the Analyze menu. The final graph was generated using GraphPad Prism 8. Each plot line represented the intensity distribution across a single seam cell from an independent animal.

For quantification of miRNA transcriptional reporter fluorescence intensity, each image stack was opened in ImageJ and a single slice with focused head was selected for further analysis. A circle ROI with a diameter of 10 pixels (1.625 µm) was drawn in the head region. For the mir-48 and mir-73 reporter, a circle was drawn on the pharynx bulb. For the mir-230 reporter, a circle was drawn on the first seam cell in the head. The intensity of the ROI was measured and presented as Mean ± SEM (t-test. *p<0.05; **p<0.01; ***p<0.001; ****p<0.0001).

## Assay of dauer formation with 1% SDS

To assay dauer formation, approximately 30 gravid adults (aged ~60 hr) were transferred to a fresh 3.5 cm NGM plate where they laid eggs for 4 hr. Afterward, all adults were removed, and the synchronized eggs were maintained at 25 °C for 50 hr. To analyze the ratio of dauer formation, animals were washed from the NGM plate and treated with 1% SDS for 20 min. Surviving animals were counted and considered as dauer, while dead animals were counted and considered as non-dauer.

## Assay of L2d formation by imaging

To assay L2d formation, approximately 30 gravid adults (aged ~60 hr) were transferred to a fresh 3.5 cm NGM plate where they laid eggs for 4 hr. Afterward, all adults were removed, and the synchronized

eggs were maintained at 20 °C for 48 hr. To analyze the morphology of animals, DIC images of animals were acquired on an OLYMPUS BX63 microscope using a 10x objective. Animals exhibiting a leaner morphology and more condensed intestinal granules were counted and considered as L2d.

## Assay of dauer formation by imaging

To assay dauer formation, synchronized fresh-hatched L1 were seeded at ~200 per 3.5 cm NGM plate and cultured at 20 °C for 70 hr. To analyze the morphology of animals, DIC images of animals were acquired on an OLYMPUS BX63 microscope using a 10x objective. Animals exhibiting a leaner and darker morphology were counted and considered as dauer.

## Quantification of animal length

To quantify animal length and analyze their growth, DIC images of animals were acquired on an OLYMPUS BX63 microscope using a 10x objective. The lengths of animals in the images were measured using the polyline tool in OLYMPUS imaging software, and the data were further analyzed in GraphPad Prism.

## Auxin treatment

To perform auxin treatment, worms were transferred to OP50-seeded NGM plates containing ~4 mM K-NAA (1-Naphthaleneacetic acid potassium salt). Briefly, fresh OP50-seeded 3.5 cm NGM plates with 3 mL NGM agar and a 25 mM K-NAA solution in ddH2O were prepared. Next, 500 μL of the 25 mM K-NAA solution was added to the NGM plates, and the lid was kept on until the solution dried. For all K-NAA treatment experiments, 500 μL of ddH2O was used as a control.

## Phylogenetic analysis of differentially expressed miRNA genes

Phylogenetic analyses were performed using the MEGA 11.0 program. First, all miRNA sequences were imported into MEGA 11.0 and aligned using ClustalX to search for conserved bases and motifs. Pairwise distance was then estimated using the default parameters, except for the bootstrap replication number parameter, which was set to 1000, and the substitution model parameter, which was set to P-distance. The phylogenetic tree for our specific miRNAs was generated using the Neighbor-Joining method and default parameters, with branches corresponding to partitions reproduced in less than 0.1 bootstrap replicates filtered out.

## Considerations on sample size and randomization

Determining sample size involves multiple factors. Typically, when testing the difference between two samples using t-tests and ANOVA, which generally require smaller sample sizes, we have found that 20–30 samples can often yield significant differentiation between the two groups. Additionally, we take into account practical constraints such as time, budget, and feasibility to avoid unnecessary costs. It's important to note that even in synchronized populations with similar starting sizes, there may be size variations over a growth period. We only exclude individuals that are clearly in an inappropriate stage, such as young L1 sizes in late L1 populations, which can occur in both mutant and control animals. The images captured are from randomized animals on slice preparations, and there is no particular order to the images when they were analyzed. Since the intensity from each image was measured using ImageJ and is devoid of human errors, we determined it unnecessary to blind the images across distinct genotypes.

## Acknowledgements

Some strains were provided by the CGC, which is funded by the NIH Office of Research Infrastructure Programs (P40 OD010440). We thank the National Bioresource Project for *C. elegans* for providing some of the strains used in this study. This work was supported by the lab start-up fund from ShanghaiTech University to YBQ; and by the National Science Foundation for Young Scientists of China (grant no. 31900397) to LW. We extend our gratitude to Shouhong Guang and Xinya Huang from the University of Science and Technology of China for their continuous contributions to the chromatin immunoprecipitation experiments. We thank Xiaoting Feng for her assistance with the MYRF-1-AID experiments and Youzhe He at BGI-Shenzhen for providing bioinformatics support. We thank the

Molecular and Cell Biology Core Facility (MCBCF) and the Molecular Imaging Core Facility (MICF) at the School of Life Science and Technology, ShanghaiTech University for providing technical support.

## Additional information

### Funding

| Funder | Grant reference number | Author |
|---|---|---|
| ShanghaiTech University | | Yingchuan B Qi |
| National Science Foundation of China | 31900397 | Lifang Wang |

The funders had no role in study design, data collection and interpretation, or the decision to submit the work for publication.

### Author contributions

Zhimin Xu, Formal analysis, Investigation, Visualization, Writing – review and editing; Zhao Wang, Investigation; Lifang Wang, Formal analysis, Funding acquisition, Investigation, Writing – review and editing; Yingchuan B Qi, Conceptualization, Funding acquisition, Project administration, Supervision, Visualization, Writing – original draft, Writing – review and editing

### Author ORCIDs
Zhimin Xu ⓘ https://orcid.org/0009-0007-6371-9345
Lifang Wang ⓘ https://orcid.org/0000-0002-6235-3069
Yingchuan B Qi ⓘ https://orcid.org/0000-0002-4267-4770

Reviewer #1 (Public review): https://doi.org/10.7554/eLife.89903.4.sa1
Reviewer #2 (Public review): https://doi.org/10.7554/eLife.89903.4.sa2
Author response https://doi.org/10.7554/eLife.89903.4.sa3

## Additional files

### Supplementary files
• MDAR checklist

### Data availability
A dataset is available at https://doi.org/10.5061/dryad.tqjq2bw71. This dataset comprises original images in CZI, TIF, or VSI formats, and datasheets in XLSX or XLS formats from quantitative and statistical analysis. The data correspond to representative images, quantitative analyses, and graphs in Figure 1A–C, Figure 2A–H, Figure 3A–B, Figure 4A–C and E, Figure 5A–F, Figure 6D–E–, Figure 7C–D, Figure 2—figure supplement 1A-B, Figure 2—figure supplement 2, Figure 2—figure supplement 3A-D, Figure 2—figure supplement 4A-B, Figure 6—figure supplement 1A-B, and Figure 6—figure supplement 2A-C. The folder and file names typically indicate essential genotype and animal stage information. The files may be supplemented by additional notes in TXT format. CZI files were acquired using a Zeiss LSM880 microscope with Airyscan. VSI or TIF files were acquired using an Olympus BX63 microscope. Excel files (XLSX/XLS) are used for organizing and presenting data summaries and results of quantitative analyses. The raw sequencing data files are available under the title 'generated from microRNA Comparative MicroRNA Sequencing Analysis in Late L1 Larvae of *C. elegans*: Wild-Type N2 Versus myrf-1(ju1121) Mutants' at NCBI Gene Expression Omnibus website with the identifier GSE262766. The plasmids and *C. elegans* strains generated in this study are available upon request from the corresponding author.

The following datasets were generated:

| Author(s) | Year | Dataset title | Dataset URL | Database and Identifier |
|---|---|---|---|---|
| Qi YB, Xu Z, Wang Z, Wang L | 2024 | Data from: Essential function of transmembrane transcription factor MYRF in promoting transcription of miRNA *lin-4* during *C. elegans* development | https://doi.org/10.5061/dryad.tqjq2bw71 | Dryad Digital Repository, 10.5061/dryad.tqjq2bw71 |
| Xu Z, Wang Z, Wang L, Qi YB | 2024 | Comparative MicroRNA Sequencing Analysis in Late L1 Larvae of *C. elegans*: Wild Type N2 Versus myrf-1(ju1121) Mutants | https://www.ncbi.nlm.nih.gov/geo/query/acc.cgi?acc=GSE262766 | NCBI Gene Expression Omnibus, GSE262766 |

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
