## [Editor Report · eLife assessment]

The microRNA *lin-4*, originally discovered in *C. elegans*, has a key role in controlling developmental timing across species, but how its expression is developmentally regulated is poorly understood. Here, the authors provide **convincing** evidence that two MYRF transcription factors are essential positive regulators of *lin-4* during early *C. elegans* larval development. These results provide **important** insight into the molecular control of developmental timing that could have significant implications for understanding these processes in more complex systems.

---

## [Referee Report · Reviewer #1 (Public review)]

In this work, the authors set out to ask whether the MYRF family of transcription factors, represented by myrf-1 and myrf-2 in *C. elegans*, have a role in the temporally controlled expression of the miRNA *lin-4*. The precisely timed onset of *lin-4* expression in the late L1 stage is known to be a critical step in the developmental timing ("heterochronic") pathway, allowing worms to move from the L1 to the L2 stage of development. Despite the importance of this step of the pathway, the mechanisms that control the onset of *lin-4* expression are not well understood.

Overall, the paper provides convincing evidence that MYRF factors have a key role in promoting *lin-4* expression in young larvae. Using state-of-the-art techniques (knock-in reporters and conditional alleles), the authors show that MYRF factors are essential for *lin-4* activation and act cell-autonomously. Results using some unusual gain-of-function alleles are supported by consistent results using other approaches. The authors also provide evidence supporting the idea that MYRF factors activate *lin-4* by directly activating its promoter. Because these results are indirect test of this, further experiments will be necessary to conclusively determine whether *lin-4* is indeed a direct target of MYRF factors. myrf-1 and myrf-2 likely function redundantly to activate *lin-4*; potential complex interactions between these two genes will be an interesting area for future work.

Overall, the paper's results are convincing. The important findings on miRNA regulation and the control of developmental timing will make this work of interest to a broad range of developmental biologists.

---

## [Referee Report · Reviewer #2 (Public review)]

Summary:

In this manuscript, the authors examine how temporal expression of the *lin-4* microRNA is transcriptionally regulated.

Comments on revised version:

In the revised manuscript, the authors have suitably addressed my original concerns.

Aims achieved: The aims of the work are now achieved.

Impact: This study shows that a single transcription factor (MYRF-1) is important for the regulation of multiple microRNAs that are expressed early in development to control developmental timing.

---

## [Author Response]

The following is the authors’ response to the previous reviews.

Thank you for your continued review and for providing insightful suggestions. Below, I share some unpublished new findings related to the MYRF ChIP, comment on the potential interplay between myrf-1 and myrf-2, and describe the modifications we've implemented to address the reviewers' comments.

(1) MYRF-1 ChIP

Our collaboration with the modERN (Model Organism Encyclopedia of Regulatory Networks) project has recently yielded MYRF ChIP data. The results demonstrate clear and consistent MYRF binding across samples, notably on the *lin-4* promoter. Given the significant detail and extensive description required to adequately present these findings, we have decided it is impractical to include them in the current paper. These results will be more suitably published in a separate ongoing study focused on MYRF's regulatory targets during larval development.

(2) Inter-regulation between myrf-1 and myrf-2

We acknowledge the interpretation that myrf-2 may act as a genetic antagonist to myrf-1, as suggested by the delayed arrest in myrf-1; myrf-2 double mutants and a trend towards increased *lin-4* expression in myrf-2 mutants. Additionally, our unpublished data suggest an elevated myrf-2 expression peak in myrf-1 null mutants during the L1-L2 transition, indicating a potential mutual repressive interaction between myrf1 and myrf-2.

On the other hand, myrf-1 and myrf-2 exhibit functional redundancy in DD synaptic rewiring and *lin-4* expression. A gain of function in myrf-2 promotes early DD synaptic rewiring. Furthermore, three independent co-immunoprecipitation analyses targeting myrf-1::gfp, myrf-2::gfp, and pan-1::gfp confirm a tight association between myrf-1 and myrf-2 in vivo. These findings challenge the notion of myrf-2 primarily antagonizing myrf-1, or vice versa.

We propose a model where myrf-1 and myrf-2 collaborate and are functionally redundant, with compensatory elevated expression when one paralog is absent. For instance, the loss of myrf-1 triggers upregulation of myrf-2, which, though insufficient on its own, accelerates the transcriptional program and exacerbates system deterioration, leading to accelerated death. How exactly this takes place is currently unclear. We notice the MYRF binding on both myrf-1 and myrf-2 genes in MYRF-ChIP.

Given the complexity of these interactions, we have chosen not to delve deeply into this discussion in the paper without more direct evidence, which would require detailed analysis.

(3) Revisions Addressing Reviewer Suggestions

(a) We have revised our interpretation of the mScarlet signal changes in myrf-1(ybq6) and myrf-2(ybq42) mutants to reflect a more nuanced understanding of their potential genetic relationship, as highlighted in the main text.

“The mScarlet signals exhibit a marked reduction in the putative null mutant myrf-1(ybq6) (Figure 1D, E). Intriguingly, in the putative null myrf-2(ybq42) mutants, there is a noticeable trend towards increased mScarlet signals, although this increase does not reach statistical significance (Figure 2C, D).”

(b) In response to feedback on Figure 2 and the characterization of *lin-4*(umn84) mutants, we've included a new series of images showing *lin-4*(umn84)/+ and *lin-4*(umn84) signals through larval stages, presented as Figure 2 Figure Supplement 2. This addition clarifies the functional status of *lin-4* nulls in our study.

“Our observations revealed that mScarlet signals were not detected early L1 larvae (Figure 2C-F; Figure 2 Figure Supplement 2).”

(c) To improve the clarity of Fig 6, we've added indicator arrows in the red, green, and merge channels, enhancing the visualization of the signals.

We appreciate the opportunity to clarify these points and hope that our revisions and additional data address the concerns raised.